# Machine learning classification of trajectories from molecular dynamics simulations of chromosome segregation

**David Geisel**, **Peter Lenz***

Department of Physics, Philipps University Marburg, Marburg, Germany

\* peter.lenz@physik.uni-marburg.de

## Abstract

In contrast to the well characterized mitotic machinery in eukaryotes it seems as if there is no universal mechanism organizing chromosome segregation in all bacteria. Apparently, some bacteria even use combinations of different segregation mechanisms such as protein machines or rely on physical forces. The identification of the relevant mechanisms is a difficult task. Here, we introduce a new machine learning approach to this problem. It is based on the analysis of trajectories of individual loci in the course of chromosomal segregation obtained by fluorescence microscopy. While machine learning approaches have already been applied successfully to trajectory classification in other areas, so far it has not been possible to use them to discriminate segregation mechanisms in bacteria. A main obstacle for this is the large number of trajectories required to train machine learning algorithms that we overcome here by using trajectories obtained from molecular dynamics simulations. We used these trajectories to train four different machine learning algorithms, two linear models and two tree-based classifiers, to discriminate segregation mechanisms and possible combinations of them. The classification was performed once using the complete trajectories as high-dimensional input vectors as well as on a set of features which were used to transform the trajectories into low-dimensional input vectors for the classifiers. Finally, we tested our classifiers on shorter trajectories with duration times comparable (or even shorter) than typical experimental trajectories and on trajectories measured with varying temporal resolutions. Our results demonstrate that machine learning algorithms are indeed capable of discriminating different segregation mechanisms in bacteria and to even resolve combinations of the mechanisms on rather short time scales.

## Introduction

A crucial task for all cellular life is to ensure proper segregation of its genetic material. Eukaryotes make use of a macromolecular machinery to deal with this task [1–3]. However, in bacteria the picture is not so clear yet [1, 4, 5]. It seems as if there are several mechanisms involved in chromosome segregation which range from purely physical forces like entropic separation of the chromosomes to protein components organizing and segregating the DNA [6–15].

**Data Availability Statement:** The trajectory data and trained models can be accessed at https://data.uni-marburg.de/handle/dataumr/135. Code for the evaluation of the data can be found in the GitHub

repository https://github.com/DavidGeisel/ML_Classification_MD_Trajectories.

**Funding:** This work was supported by the Deutsche Forschungsgemeinschaft (DFG, TRR174). The funders had no role in study design, data collection and analysis, decision to publish, or preparation of the manuscript.

**Competing interests:** The authors have declared that no competing interests exist.

At the center of many of the processes and mechanisms that regulate the organization and segregation of DNA in the cell is the origin of replication (*ori*) [2, 9, 16]. The *ori* is central to the function of two of the most frequently mentioned proteins associated with segregation, namely the ParAB system and the structural maintenance of chromosomes (SMC) complex [2, 6, 9, 17]. Both proteins are loaded at or near the *ori*. Thus, the movement of the *ori* through the cell is of special interest if one wants to understand segregation of DNA in bacterial cells [1, 2, 4, 6]. Recent advances in imaging technologies and single-particle tracking (SPT) have made it possible to extract trajectories of individual chromosomal loci throughout the cell [4, 16, 18, 19]. In particular, the *ori*-region can be tagged with fluorescent dye particles and illumination with lasers enables tracking of the movement of the molecule. Given this experimental progress, the question arises if it is possible to extract information about the underlying cellular processes from the trajectories of single molecules and in particular from the trajectory of the *ori* during the segregation process. A new approach to this problem has recently come up with the application of machine learning (ML) algorithms to the classification of single trajectories [20–22]. So far, these approaches have proven useful in the discrimination of diffusion modes. To be able to do this, synthetic data are produced by computer models on which the machine learning algorithms can be trained. This is necessary as training of machine learning algorithms requires large sets of data. As input for training one can either use complete trajectories (as high-dimensional input vectors) or extracted statistical features of the trajectories (as low-dimensional input vectors). In the latter case training is less time consuming and less demanding on computational resources [22–25].

In this work, we used Molecular dynamics (MD) simulations to investigate the segregation of chromosomes in the cell. To do so, we extended our model of entropic segregation of chromosomes [4] by implementing the action of the ParAB system and SMC as additional key players orchestrating the separation of DNA in bacteria. By switching the two proteins on and off we simulated knock-out mutants. We further used two different schemes of replication, the track model and the factory model. In this way, we created eight different scenarios of chromosome segregation in a bacterial cell (two replication models times four segregation mechanisms) from which we extracted the trajectories of the newly duplicated *ori*. Thereby, we simulated typical single-particle tracking experiments with the advantage of being able to achieve a much better time resolution than achievable in *in vivo* experiments. We then trained our machine learning algorithms to classify the trajectories. We used two linear models, a logistic regression classifier (LR) and a support vector machine (SVM) and two tree-based classifiers, a random forest (RF) and a gradient boosting (GB) algorithm. As input for our classifiers we either presented complete (normalized) trajectories or a sample of 8 statistical features to reduce the input dimension. Finally, we tested our classifiers on shorter trajectories of only some seconds to see if they are also able to classify data as might result from experiments.

This paper is organized as follows. First, we give a brief overview on the mechanisms for replication and segregation in bacteria which we implemented in our analyses. We then present our MD framework and the classification models used in this study. Furthermore, the statistical features used to describe the trajectories are discussed. We then compare the classification results of the two approaches using high-dimensional and low-dimensional input vectors. Furthermore, the classification accuracies for short trajectories and trajectories with varying temporal resolution are presented. The various results are discussed and interpreted in the final section.

## Biological background and simulation methods

In bacterial cells, replication and segregation of the chromosome occur simultaneously [1, 7]. The spatiotemporal coordination of these two processes is a challenging task, for which bacteria have evolved different mechanisms. In the following, we briefly summarize these replication and segregation mechanisms that have been addressed in this work. For a broader review of relevant mechanisms, see, e.g., Refs. [1, 4, 5, 7].

### Replication and segregation mechanism in bacteria

**Replication schemes.** Bacteria and eukaryotes share common features in DNA replication [26, 27]. For example, replication is always initiated at a defined structure on the DNA, the origin of replication (*ori*). The *ori* regions are specified by particular DNA sequences which attract initiator proteins. These proteins are responsible for the formation of a nucleoprotein complex, the replisome, which ultimately initiates DNA replication [28]. As its shape resembles the letter Y the localized region of replication moving along the DNA is called a replication fork. Since bacterial chromosomes are circular, their duplication occurs bidirectionally. That is, two replisomes duplicate DNA in opposite directions, starting at the *ori*. Replication is finished when the two replication forks meet in the termination region (*ter*) [26–29].

Nevertheless, there are still many unanswered questions. For example, it is unclear if the replisomes duplicating the chromosome are fixed in bacterial cells. Two opposing models are discussed in the literature, the factory model and the track model [15, 29–32]. In the factory model, it is assumed that the replisomes are fixed in the center of the cell and thus form a 'replication factory' there. Consequently, the replisomes are attached to each other and the mother chromosome is pulled through the replication factory while being duplicated. Such a model might require an additional mechanism in order to localize the replisomes at the cell center. It was proposed that this mechanism could be provided by protein complexes organizing the newly synthesized DNA behind the replication and thereby immobilizing the replisomes. A possible advantage of this mechanism is that the replisomes are prevented from coiling along the DNA and interweaving the newly replicated strands [32]. In contrast, the track model assumes that the replisomes are separable and mobile. According to the track model the replisomes would move along the chromosome in opposite directions like trains on a track while duplicating the DNA. Consequently, no anchoring mechanism is needed for the track model. Instead, the spatial organization of the chromosome determines the movement of the replisomes [15, 29–32].

In the literature arguments in favor of both replication schemes can be found. For *Bacillus subtilis* (*B. subtilis*) the existence of a replication factory was postulated quite early on [30, 31]. More recently, Mangiameli et al. [32] performed time-lapse microscopy in *B. subtilis* and *E. coli* and observed that the replisomes reside in close proximity during large parts of the replication period. On the other hand, Japaridze et al. reported independently moving replisomes in *E. coli* [15]. To resolve these apparently contradicting observations it was suggested that the replisomes might not be strictly connected by a physical link, but that there might be a confinement to a specific subcellular region, e.g., as a consequence of the dynamics of the growing nucleoid [29]. Finally, the question arises whether one of the two possible replication models offers advantages in the segregation of the chromosomes. For the factory model, it was suggested that chromosome segregation might be facilitated by pushing the DNA to the cell poles after replication at the center of the cell [32]. To address this question, we will first review some of the major segregation mechanisms for bacterial chromosomes in the following section. Snapshots of our implementation of the track and factory model in our molecular dynamics (MD) simulations can be found in [4].

**Segregation mechanisms.** All cellular life is faced with the challenge of ensuring that its genetic material is segregated reliably. While the mechanism of chromosome segregation in eukaryotes is managed by a well-understood macromolecular machine, the mitotic spindle, there is no such unique mechanism in bacteria [1–5]. Instead, we are just beginning to understand the mechanisms responsible for chromosome segregation in bacteria. They involve both specific protein components and other, mechanical-based mechanisms [6–15]. Of central importance for both segregation and replication of the bacterial chromosome is a specific chromosomal locus, the origin of replication (*ori*) where the bidirectional chromosome replication starts. However, the *ori's* movement patterns through the cell are not fully understood yet [1, 2, 6]. It is believed that the *ori* undergoes simultaneous active and thermally driven transport in the cell. In the following sections, we present three of the most prominent mechanisms proposed to play a crucial role in bacterial chromosome segregation and segregation of the *ori* regions in particular.

One theory for segregation is based on the elastic properties of chromosomes. Starting point was the observation that confined polymers can actively segregate while disconnected, physically identical particles tend to mix [12, 13]. The reason for this is that intermingled long polymers have less conformational entropy then completely separated ones. Therefore, entropic forces act on the intermingled polymers and provide a segregation force. A key aspect for this mechanism to work is the role of confinement. Within confinement the polymers behave like loaded entropic springs storing the free energy produced by the DNA-protein interaction [13, 33–35]. Experimental evidence for this effect was recently presented for *E. coli* [15]. In this study, cells with an increased width were investigated and it was shown that the probability of successful segregation of chromosomes decreases with increasing cell width, pointing out confinement as an important driver of DNA segregation [15]. A number of other studies have demonstrated the importance of entropic effects in the organization and segregation of bacterial chromosomes. The entropic effects cause a tendency of chromosomes to adopt a conformation that maximizes the available degrees of freedom for their segments. This tendency both leads to segregation of two chromosomes from each other, as well as to a compactification of individual chromosomes in the presence of a large number of additional crowding particles. In the latter case, the loss of conformational freedom of the polymer as a result of its compaction is overcompensated by additional available volume for the crowding particles [2, 7, 33, 36–38].

As mentioned, this entropic separation of chromosomes is a purely physical effect based entirely on the elastic properties of chromosomes and fundamental physical laws. Therefore, the entropic force is certainly one of the key factors for chromosome segregation. However, there are also numerous other experimental indications for further key factors and mechanisms for successful segregation of the duplicated DNA in different model organisms. It is believed that a variety of mechanisms with both facilitating and prohibiting effects have an influence on chromosome partitioning [6].

One mechanism that is used in bacteria to achieve accurate chromosome segregation is the use of protein machines that interact with specific sites on chromosomes. In bacteria, these sites are known as partitioning (*par*) loci [39]. Interestingly, although these systems are thought to act similarly in all bacteria, their contribution to origin segregation varies dramatically [8]. One of these protein complexes is the widely conserved ParAB system which plays a major role in bacterial chromosome segregation [6, 40].

In *C. crescentus*, the ParAB system is critical for segregating origins from one pole to the other. Induction of dominant negative allele of ParA was shown to dramatically block origin segregation [41]. However, *B. subtilis* mutants lacking ParA faithfully segregate their chromosomes as assayed by the production of anucleate cells [42]. Nevertheless, the ParAB system

plays a central role in segregating origins towards the cell poles in many bacteria and is the closest analog to a eukaryotic-like mitotic apparatus [8]. Visualization of origin dynamics by time-lapse microscopy in the absence of ParA indicate that the ParAB system facilitates the directed movement of the origins to the cell poles and helps to establish and maintain the *ori-ter* pattern of the newly replicated DNA in *B. subtilis* [43].

The ParAB-system consists of three components: the DNA sequence *parS*, the DNA-binding protein *ParB*, and the deviant Walker A-type ATPase ParA [6]. Typically, *parS* sequences are found near the *ori* in many bacteria. ParB recognizes these sequences and binds to them. Thereafter, ParB is thought to spread on flanking sequences to form the so-called ParB/*parS* partition complex. The last component, ParA, dimerizes upon ATP binding, which in turn promotes nonspecific DNA binding [6]. Altogether, the ParAB system appears to "pull" the duplicated origin region to the opposite cell pole, where it is anchored similarly to its sibling that remains at the other cell pole [4, 6, 40, 44]. An important question within the mechanism of ParAB-dependent transport concerns the origin of the translocation force. Lim et al. propose a model, where this force is derived from the elastic property of the chromosome [6]. In accordance with the proposed model of Wiggins et al. [45] that the bacterial chromosomes behave like elastic filaments, Lim et al. assume a DNA-relay mechanism, in which the DNA-associated ParA-ATP dimers serve as transient tethers that harness the intrinsic dynamics of the chromosome to relay the partition complex from one DNA region to another [6]. To estimate the characteristic elastic force generated from the chromosomal locus dynamics, Lim et al. tracked single loci positions prior to replication and segregation. From a Gaussian fit to the distributions they derived as estimate for the characteristic force $F \approx 0.06 pN$ [6].

A further group of key proteins for chromosome dynamics are SMC proteins [7, 10, 17, 39, 46]. They are essential proteins which are conserved throughout all domains of life and SMC condensin complexes play a crucial role in constraining DNA in both eukaryotes and prokaryotes [47]. They are also crucial for correct chromosome compaction and segregation [8, 47]. The bacterial SMC complex (MukBEF complex in *E. coli*) consists of the SMC (*MukB*) protein, *ScpA* (*MukE*), and *ScpB* (*MukF*). The SMC proteins are characterized by interwined coiled-coil domains that have hinges at both ends enabling them to embrace DNA strands dynamically [39].

In eukaryotes, condensins act at the earliest stages of mitosis to compact and resolve interphase chromosomes into rod-shaped structures that assemble at the metaphase plate [46]. In bacteria it was found that condensins constrain and bridge DNA segments and topologically embrace DNA helices [10]. Wang et al. proposed a model of SMC in which the ring-shaped complexes encircle the DNA flanking their loading site, tethering the DNA duplexes together [17]. Furthermore, they showed that the SMC complex plays an important role in the formation of topologically associated domains (TADs) [17]. Another important function of SMC proteins in bacterial cells is their involvement in DNA segregation. It was found that origin segregation in *B. subtilis* is a task shared by the condensin complex and the ParAB partitioning system [9]. Here, condensin resolves replicated origins by promoting the juxtaposition of DNA flanking *parS* sites, drawing sister origins in on themselves and away from each other [10, 17]. For this purpose in *B. subtilis* condensin is loaded at centromeric *parS* sites, where it encircles DNA and individualizes newly replicated origins [10, 17]. However, while SMC is loaded at the origin it still is present and acts along the complete chromosome arms [10].

For *B. subtilis* it was shown that rapid inactivation of SMC during fast growth leads to a failure in resolving newly replicated origins and a block of chromosome segregation [9]. On the other hand, during slow growth the ParAB partitioning system showed to provide enough origin segregation activity to support chromosome segregation in the absence of SMC [9]. Nevertheless, slow growing *B. subtilis* cells lacking SMC have more heterogeneous nucleoid

morphologies, in part due to a defect in resolution of replicated origins [8]. Also in slow grow-ing *E. coli*, nucleated cells that lack condensin adopt an *ori-ter* configuration rather than their typical left-*ori*-right [8].

The importance of SMC proteins in chromosome dynamics was also demonstrated in com-puter simulations. In polymer simulations of chromosome dynamics it was shown that a single mechanism of loop extrusion by condensins can robustly compact, segregate and disentangle chromosomes [48]. In these simulations eukaryotic chromosomes were implemented as flexi-ble polymers where each condensin complex was modeled as a dynamic bond between a pair of monomers [48]. Furthermore, it was shown that a limited number of $\sim 30$ condensin com-plexes per replication origin can organize DNA in *B. subtilis* [17].

## Molecular dynamics of chromosome segregation

We used Molecular dynamics (MD) simulations to simulate the segregation of chromosomes in the cell. MD simulation is one of the most widely used techniques in computational chemis-try. The reason is its (relative) simplicity and ability to accurately sample the conformational space of a molecular system [49, 50]. The main idea of MD simulations is to integrate Newton's equations of motion [see Eq (1)] for the molecular constituents of the system of interest, in our case the building blocks of a biopolymer, and thereby evaluate the time-dependent behavior and evolution of a molecular system as it samples conformational space [49]:

$$m_\alpha \ddot{\vec{r}}_\alpha = -\frac{\partial}{\partial \vec{r}_\alpha} U_{total}(\vec{r}_1, \vec{r}_2, \ldots, \vec{r}_N) \ , \ \alpha = 1, 2, \ldots, N. \tag{1}$$

Here, $m_\alpha$ represents the mass of constituent $\alpha$, $\vec{r}_\alpha$ is its position, $\ddot{\vec{r}}$ is the twofold time derivative of $\vec{r}$ (= acceleration), and $U_{total}$ the potential energy that depends on all constituents positions and, thereby, couples the motion of all degrees of freedom [50]. MD simulations have already provided detailed information on local motions and conformational changes of proteins and DNA [49].

For our simulations we used the software package `ESPResSo` [51] and expanded our model introduced in [4]. Within this model, the bacterial chromosome is represented by a chain of spherical beads. Each bead represents a unit of compacted DNA (a detailed descrip-tion of the used model for DNA can be found in S1 Appendix). The simulations start with a circular chromosome which is replicated in the course of the simulation. We constructed the starting configurations in a separate Monte Carlo simulation according to the scheme pre-sented in [34]. Here, the cellular volume is discretized and represented by a cubic lattice. The chromosome configuration then corresponds to a random walk on the lattice where each step represents a bead of DNA. The resulting configuration can then be used in the MD simulation. Interactions among chromosomes and with the cell wall are defined by specific potentials (for a detailed description of the interactions see S2 Appendix). Before the start of the actual simu-lations, an equilibration phase was set up to prevent unphysical forces from arising between closely spaced beads. In this 'warm-up' phase, the forces occurring in the simulation were arti-ficially limited to a certain cap value which was then gradually increased before the full poten-tial was active at the start of the actual simulation. Thus, initial overlaps of individual beads were avoided and an equilibrated start configuration was ensured. More details on this can be found in our previous study [4]. In the next section, we describe the implementation of the above-mentioned mechanisms of replication and segregation of bacterial chromosomes in our MD simulations.

**Implementation of replication schemes.** In bacteria, replication and segregation occur simultaneously. While the replisomes duplicate the chromosome bi-directionally, the daughter

chromosomes already start to separate [1, 5, 29]. For the implementation of this process, the replication in our simulations was divided into duplication steps in which always two beads are duplicated by the replication polymerases (one bead in each direction) to account for the bidirectional replication of the chromosome [4]. Between the single duplication events the partially replicated chromosomes exhibit thermal fluctuations and already start to segregate. For the implementation of the track model, the new beads were created randomly around the original position of the old bead to be duplicated. Thus, the replisomes migrated along the old chromosome. In contrast, the replisomes are fixed at midcell within the factory model. Therefore, we added an additional bead in the center of the cell to serve as a replication factory. Subsequently, the mother chromosome was connected to the replication factory. For this purpose, the two beads closest to the replication factory were connected to it by additional harmonic springs. During each replication step, those two beads were duplicated and then the subsequent beads were reconnected to the replication factory with harmonic bonds. Between the individual replication steps, the beads of the parent chromosome to be duplicated in the following were thus moved to the replication factory by the harmonic spring force, simulating the pulling effect of the factory model. Thus, localization of replication in the center of the cell could be implemented.

In typical model organisms such as *E. coli* and *B. subtilis*, the process of replication takes between 20–60 min [4, 5, 52]. As a result, direct simulation with MD techniques is not feasible due to the enormous computation time. Similar problems can be found in many applications of MD simulations, where the dynamic evolution of a biological system cannot be represented in its entirety by simulations, but where ways to accelerate the simulation time are needed [49, 53, 54]. In our model of simultaneous replication and segregation of a bacterial chromosome, it was possible to speed up the simulation time because we found a clear separation of time scales: replication takes much longer than segregation. Thus, the duration of the combined process of replication and segregation is determined by the speed of the replisomes along the chromosome. In contrast, the simultaneous separation of the duplicated material is very efficient and fast. These findings were noted in our earlier study [4] and are summarized again in S3 Appendix.

**Implementation of segregation mechanisms.** For the implementation of the SMC molecules, we used the results of [17], where it was found that a limited number of $\sim 30$ condensin complexes per replication origin can organize DNA in *B. subtilis*. We, modeled condensin complexes as dynamic bonds between pairs of monomers as it was suggested in [48]. Consequently, in our simulations, opposite beads on the two chromosome arms are connected by an additional "SMC bond". For this the same harmonic spring potential was used as for the connection of the beads to each other and to the replication factory. This simplified approach has also been used successfully in [48] to model the linkage of chromosome arms by SMC. We keep the idea that SMC is loaded at the *ori*-region by successively connecting the beads following the *ori* in the replication with bonds. An example snapshot of a chromosome with and without SMC in our simulations is shown in Fig 1.

In order to implement the ParAB system into our simulations, we needed to derive the value of the force by which it pulls the duplicated *ori* towards the cell pole. To do so, we followed the procedure from Lim et al. [6] to get an estimate of the elastic force resulting from the dynamics of the singular loci of the chromosome. As described above, the DNA-relay model of Lim et al. suggests that the translocation force of the ParAB complex results directly from the intrinsic elastic properties of the chromosome. It is assumed that the DNA-associated ParA-ATP dimers serve as transient tethers that harness the intrinsic dynamics of the chromosome to relay the partition complex from one DNA region to another [6]. Consequently, Lim et al. were able to estimate the expected elastic force of the ParAB complex by tracking the

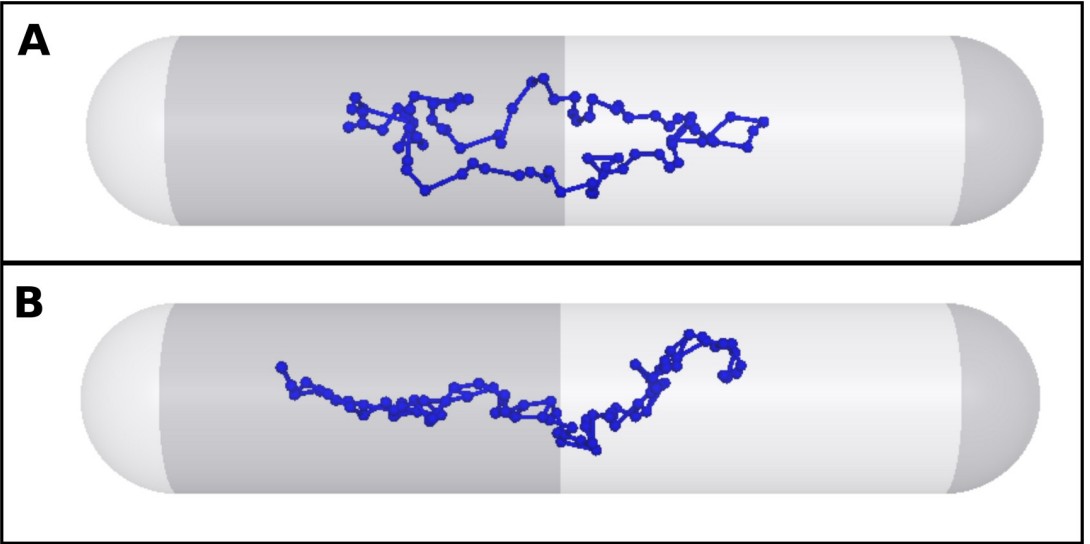

**Fig 1. SMC implementation.** Example snapshots of our simulations for a chromosome without (upper picture A) and with (lower picture B) SMC bonds.

positions of individual loci before the onset of replication and segregation. The same approach can be used in our MD framework to obtain an estimate of the force of the ParAB system within the simulations. Therefore, we analyzed the step size distributions of single chromosomal loci in the cell prior to replication and segregation in our simulations. Thereby, the probability $P(\triangle x)$ of a locus to fluctuate around its equilibrium point is given by

$$P(\triangle x) \sim \exp^{-\frac{E(\triangle x)}{k_B T}}, \tag{2}$$

where $k_B$ is the Boltzmann constant, $T$ is the absolute temperature, and $E(\triangle x)$ is the energy associated with the fluctuation [7]. For an elastic force with spring constant $k_{sp}$ and displacement $\triangle x$ we can write $F = k_{sp} \triangle x$. Here, the energy becomes $E(\triangle x) = k_{sp} \triangle x^2 / 2$ and the probability distribution can be written as

$$P(\triangle x) \sim \exp^{-\frac{k_{sp} \triangle x^2}{2 k_B T}}. \tag{3}$$

The probability distribution in Eq 3 represents a Gaussian distribution of the form $f(x) \sim \exp^{-\frac{1}{2}\frac{\triangle x^2}{\sigma^2}}$ [7]. Thus, we obtain the standard deviation $\sigma$ from

$$\frac{1}{\sigma^2} = \frac{k_{sp}}{k_B T}. \tag{4}$$

Consequently, the elastic force exerted by the ParAB system on the *ori* can be obtained from a Gaussian fit to the step size distribution as $F = k_{sp} \sigma$. Fig 2 shows such a fit to our simulation data.

From the fit in Fig 2 we obtained $\sigma = 72.7$nm. Using this, the elastic force (for a temperature of 300K) becomes

$$F = k_{sp}\sigma = \frac{k_B T}{\sigma} \approx 0.057 \text{pN}. \tag{5}$$

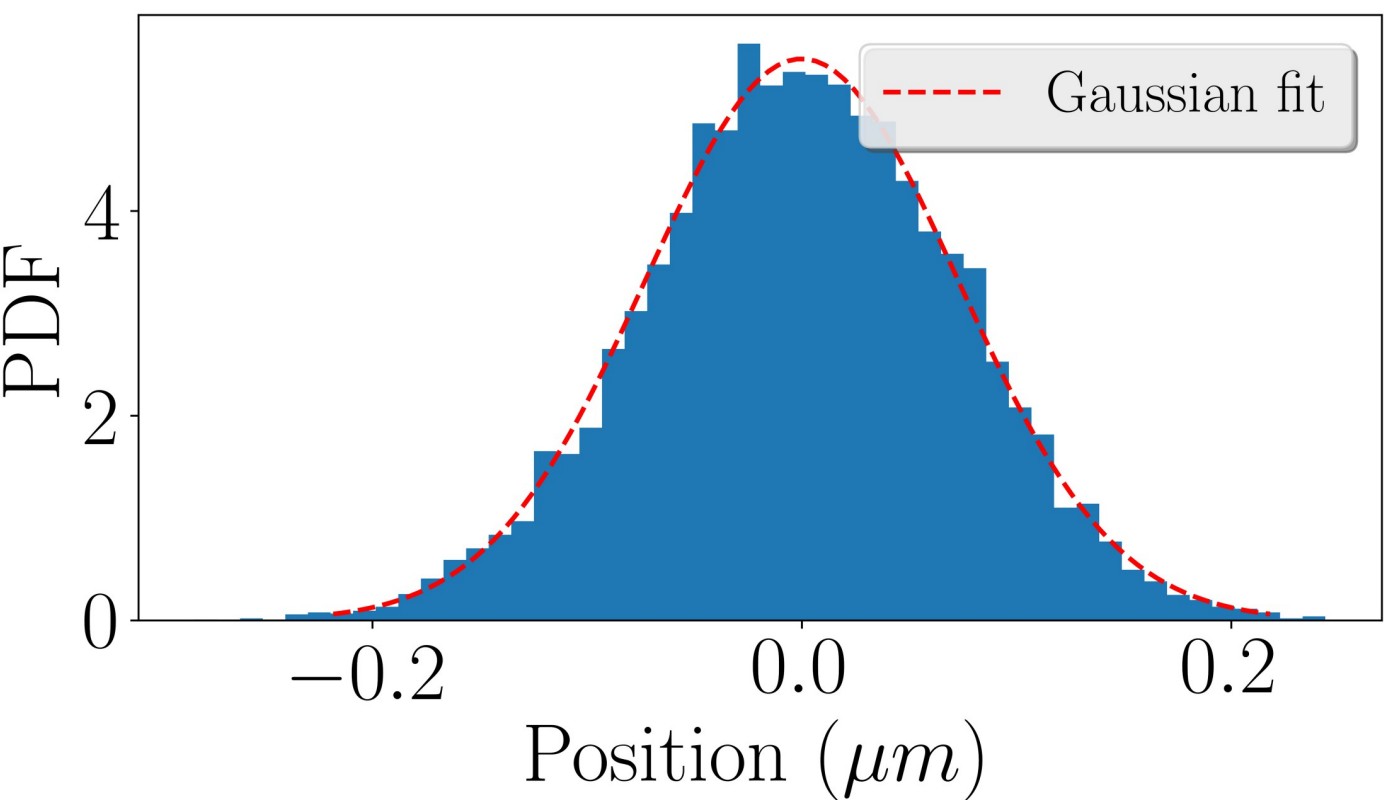

**Fig 2. Derivation of the pulling force of the ParAB system.** Step size distribution of a single loci from a freely diffusing chromosome in the cell. The histogram shows a probability density (PDF) where each bin displays the bin's raw count divided by the total number of counts and the bin width. The red line is a Gaussian fit to the data shown in blue. The standard deviation of the fitted Gaussian is 72.7nm.

This is almost the same value as the one found by Lim et al. of $F = 0.06$pN [6]. We implemented this force in our simulations as a constant external force pulling the newly duplicated *ori* towards the cell pole.

### Machine learning approach

In this section we present the machine learning (ML) workflow and the ML models used to classify the trajectories obtained by the MD simulations described above. In Fig 3 a schematic representation of the workflow applied in this study is shown.

As depicted in Fig 3 we split our complete data into separate training and test sets. The training data is used to determine optimal model architectures (hyperparameter tuning, see S5 Appendix for further information) and to then train the ML model with these architectures. Evaluation of the prediction accuracies of the ML models then follows on data from the test sets that the models have never seen before. We chose a splitting of 70% training data to 30% test data for this procedure. In order to ensure that the determined accuracies of the classifiers are not dependent on a single random split in training and test data, five-fold nested cross-validation [55, 56] was used to check the results of the single split into train and test data (see S4 Appendix for a brief summary of this approach). In the following section, a brief description is given of the ML models used.

**ML models.** In this study we used four different classifiers and compared their classification accuracies. Two of the used classifiers are ensemble methods based on decision trees. The

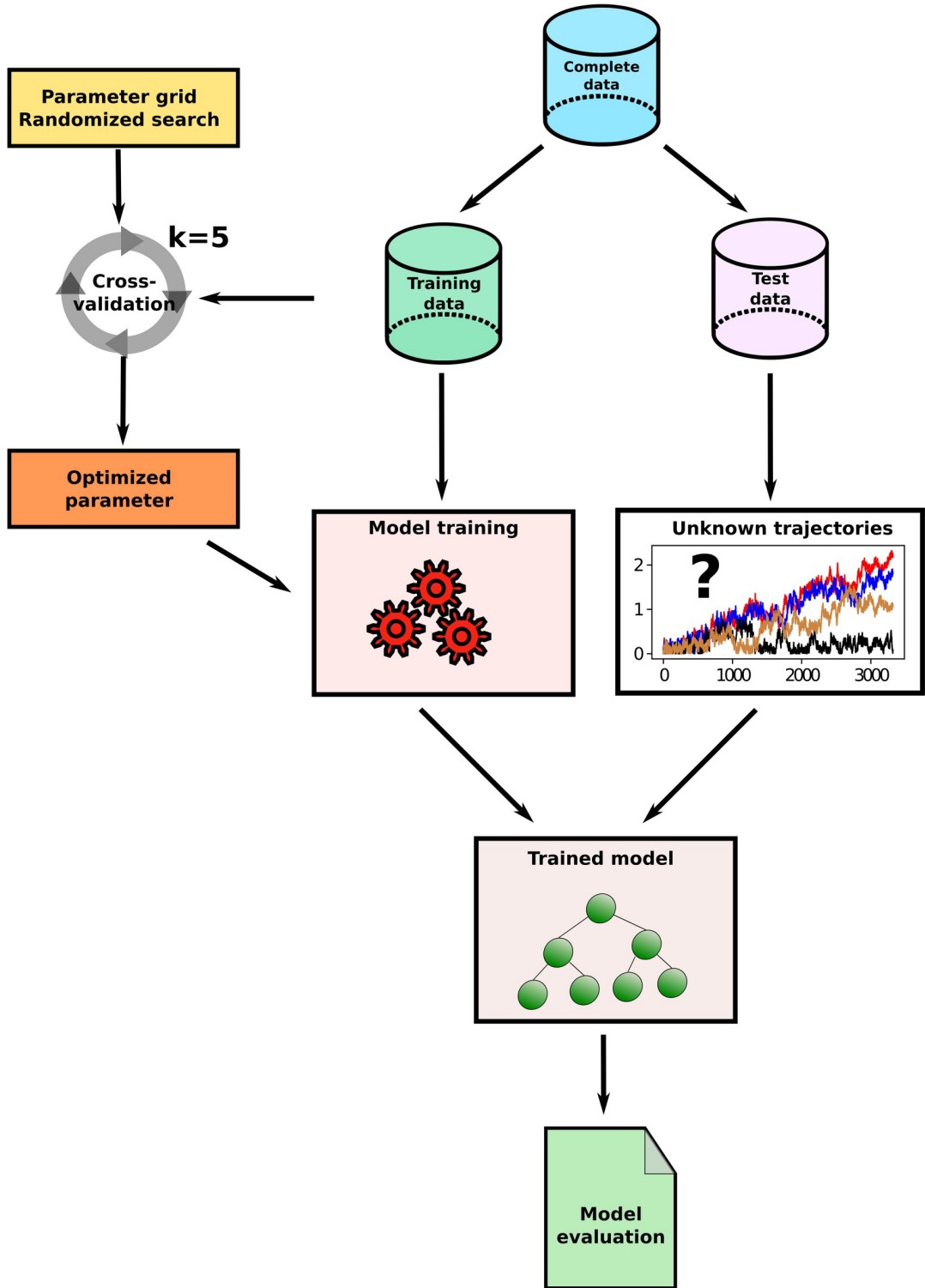

**Fig 3. Schematic overview of the ML workflow.** The complete data is divided into training and test sets. The training data is used for parameter optimization and subsequent model training. The final model evaluation is performed on the test data which has never been seen before by the model.

first one is a random forest classifier (RF) and the second one is a gradient boosting classifier (GB). The remaining two classifiers belong to the group of linear models. In such models the target value is expected to be a linear combination of the features. We used a support vector machine (SVM) and a logistic regression classifier (LR). All models were implemented using the `scikit-learn` library from python.

Both the RF and GB classifier are so-called ensemble learning methods, i.e. methods that generate many classifiers and aggregate their results. In both cases the basic classifier is a decision tree [24]. A decision tree produces recursive binary splits of the input space, so that samples belonging to the same label are grouped together [21]. The input is also called the root node of the tree. The subsets created by splitting are called successor children. One speaks of a terminal node if no further splitting of the resulting subsets is possible or if all samples at a node belong to the same class (i.e. the node is pure). The final output is obtained at the terminal nodes [20, 21]. There are two commonly used criteria that are used to perform the splits. These are the Gini impurity and the information gain [20, 57, 58]. The Gini impurity quantifies how often a randomly selected element from the set would be mislabeled if it were randomly labeled according to the distribution of labels in that set. It can be written as

$$I_G = \sum_{i=1}^{J} p_i(1 - p_i),\tag{6}$$

where $p_i$ is the fraction of items labeled with class $i$ in the set and $J$ is the number of classes [20].

In order to calculate the information gain of a split, one measures the reduction of information entropy by calculating the difference between the entropy of a parent node and the weighted sum of entropies of its children nodes [20]. A final decision tree classifies unseen data by passing it through the nodes of the tree, where each decision is made with respect to which direction to take [21]. While decision trees are easy to interpret and do not require data processing, they still have a number of disadvantages. One such disadvantage is the tendency of decision trees to overfit data. Furthermore, small variations in the data may lead to a completely different tree. For this reason, nowadays ensembles of decision trees are mainly used, in which several decision trees are combined. With this approach, it is possible to eliminate many of the disadvantages and achieve better results [20, 21, 24, 57].

One such ensemble classifier is the random forest classifier (RF). The basic idea behind a RF is called bagging. In this approach multiple training data sets are constructed by bootstrapping from the complete training set. Subsequently, many decision trees are trained on the bootstrapped data. The RF then combines the output of the individual trees by averaging their predictions [21, 24, 57, 59–61]. The final prediction $\hat{f}_{bag}(x)$ of a RF can be written as

$$\hat{f}_{bag}(x) = \frac{1}{B} \sum_{b=1}^{B} \hat{f}^b(x),\tag{7}$$

where $B$ is the number of different bootstrapped training data sets and $\hat{f}^b(x)$ is the prediction of the $b$-th decision tree. Another important feature in the construction of a RF is that for each split of a decision tree in the RF, only a random subset of the features is used. This method, proposed by Breiman [60], prevents strong correlations between the individual decision trees in the ensemble. Altogether, the bias of the ensemble is slightly higher than that of a singe decision tree, but the model has lower variance and is more robust to variations in the dataset [21].

While the single decision trees in a random forest are independent of each other, in the gradient boosting (GB) method the trees are built sequentially by learning from mistakes made by the ensemble [24]. This process is called boosting. In an iterative approach every new decision

tree is trained with respect to the error of the ensemble learnt so far and added to the ensemble afterwards. This process improves the prediction accuracy of the ensemble in areas where it did not perform well before [57, 58, 61–64].

In addition to the two tree-based models, we also used two linear classifiers. In linear models one expects the target value, $y(\vec{\beta}, \vec{x})$, to be a linear combination of the features, $x_i$ [58]. The features are weighted with the coefficients $\beta_i$, so that a linear model can be written as

$$y(\vec{\beta}, \vec{x}) = \beta_0 + \beta_1 x_1 + \ldots + \beta_n x_n. \tag{8}$$

The first linear model that we used is the logistic regression (LR) classifier. It is one of the most commonly used linear models for supervised classification. In its typical form, the LR classifier is a binary classifier, predicting the probability that an observation is part of a given class (class 1 while otherwise class 0) [65, 66]. To achieve this, the general equation of a linear model as noted in Eq 8 is combined with a logistic function which maps an input $z$ as $f_{log}(z) = \frac{1}{1+e^{-z}}$. With this, the output values are limited to the range between 0 and 1 and can be interpreted as probabilities. The probability for the occurance of an event is then written as

$$P(y(\vec{\beta}, \vec{x})) = \frac{1}{1 + e^{-y(\vec{\beta}, \vec{x})}}. \tag{9}$$

For probabilities greater than 0.5 the classifier predicts that the observation belongs to class 1. For values below 0.5 class 0 is predicted [65, 66]. To handle a multiclass problem with the LR classifier, one can replace the logistic function with a softmax function (multinomial logistic regression). Alternatively, one can apply the one-vs-rest scheme (OVR) in which a separate model is trained for each class [65, 66]. The coefficients $\beta_0, \beta_1, \ldots, \beta_n$ are estimated using maximum likelihood. In this optimization procedure one tries to maximize the likelihood that the observed values of the dependent variable may be predicted from the observed values of the independent variables [57, 58]. Furthermore, regularization is applied to reduce variance of the trained model. For this task a penalty term is added to the loss function penalizing complex models [58]. In our work we used the `scikit-learn` implementation of the LR classifier with the L2 penalty and the "lbfgs" optimization algorithm.

A support vector machine (SVM) was chosen as the second linear classifier. The approach of a SVM is to classify data by finding a hyperplane that divides the data into the respective classes. One defines the margin of a SVM as the distance from the hyperplane to the nearest expression vector. The goal of a SVM classifier is to maximize this margin between the classes [57, 58]. However, as data in most classification problems are not linearly separable, the SVM must find a balance between finding the maximum margin and misclassifying as little data as possible. To control this task, one introduces a penalty $C$ to be applied to classification errors. For a small penalty $C$ the classifier will have lower variance at the cost of a bigger bias compared to high values of $C$ [57, 58, 67]. Another central component of an SVM classifier is the kernel function. With the kernel function one can transform non-separable input data into a higher-dimensional space, in which the classification problem can be solved [67]. With the definition of the kernel, an SVM classifier can be described with the following formula.

$$f(x) = \beta_0 + \sum_{i \in S} \alpha_i K(x_i, x_{i'}). \tag{10}$$

Here, $\beta_0$ is the bias and $S$ is the set of all "support vector observations", i.e. data points close to the hyperplane which are included in the decision function of the optimizer. $\alpha$ denotes the model parameters to be learned. The kernel function $K$ compares the similarity between two

support vector observations, $x_i$ and $x_{i'}$ [57, 67]. In this work we used a linear kernel given by

$$K(x_i, x_{i'}) = \sum_{j=1}^{p} x_{ij} x_{i'j}.$$ (11)

Here, the number of features is denoted by $p$.

In the following section a description of the feature design that we have chosen in this work is given.

**Feature design.** The classifiers presented in the previous section each require an input vector in which the original data is grouped in the form of features and made processable for the classifier. The process of extracting such features from the raw data is also called feature engineering or feature design and has a great influence on the achieved classification accuracy [57, 58, 61]. In this work, we used two different approaches for feature design, comparing an approach with a high-dimensional input vecor and an approach with a low-dimensional input vector.

In the first approach we have used the preprocessing protocol suggested by Muñoz *et al.* [21]. The idea of this approach is to construct rescaled trajectories via the normalized displacements of the original trajectories. This results in comparable magnitudes of the rescaled trajectories and allows classification of heterogenous data from various spatiotemporal scales. Details of this process can be found in [21]. In the following, the rescaled trajectories can be used as high-dimensional input vectors for the classifiers, so that the positions along the trajectories correspond to the features. Using this method, Muñoz *et al.* were able to achieve very good classification accuracies with a RF classifier for trajectories of different diffusion models and confirm this also for particularly short trajectories consisting of few data points [21].

In a second approach we created low-dimensional input vectors from a set of statistical features that were calculated from the original trajectories and designed specifically for this purpose. In doing so, one hopes to reduce overfitting of the classifiers and reduce the computational cost for the classifiers. Furthermore, it might be possible to connect statistical features of the trajectories with the mechanisms producing them and thereby deepen the understanding of the mechanisms. Suggestions for statistical features of the trajectories can be found in the literature, e.g. for quantitative comparison of trajectories from single particle tracking (SPT) experiments [20, 22–25]. For our study, we selected eight features, which are described in the following.

The mean squared displacement (MSD) of a trajectory is commonly used to differentiate various types of diffusion. For a trajectory of $N$ positions $x_i (i = 1, \ldots, N)$ it can be written as

$$\mathrm{MSD} = \langle r_n^2 \rangle = \frac{1}{N-n} \sum_{i=1}^{N-n} |x_{i+n} - x_i|^2.$$ (12)

Here, the index $n$ defines the lag time. This is the time step considered in the evaluation of the MSD along the trajectory. For example, for $n = 2$, the formula evaluates the MSD of the trajectory between every point and its evolution after two time steps. To calculate our first feature (MSD), we set $n = 1$. Variation of $n$ to $n = 1, \ldots, N - 1$ allows the computation of MSD curves for different lag times in the following, from which further features (exponent alpha, mean squared displacement ratio) can be obtained as described below.

Typically, a diffusive motion is also characterized by the following relation:

$$\langle r_n^2 \rangle = 4D(n \triangle t)^\alpha.$$ (13)

Here, $D$ is the diffusion coefficient, $\triangle t$ is the elapsed time and $\alpha$ is the anomalous exponent. If

$\alpha < 1$ the motion is characterized as anomalous diffusion while $\alpha \approx 1$ for normal diffusion [22]. We used $\alpha$ (Alpha) obtained from a fit to the MSD curves as our second feature.

As our third feature we used the mean squared displacement ratio (MSDR) defined as

$$\langle r^2 \rangle_{n_1, n_2} = \frac{\langle r^2_{n_1} \rangle}{\langle r^2_{n_2} \rangle} - \frac{n_1}{n_2} \quad \text{with } n_1 < n_2. \tag{14}$$

The indices $n_1$ and $n_2$ define the different positions at which the MSD curve is evaluated. Thereby, the MSDR characterizes the shape of the MSD curve and can be calculated by setting $n_2 = n_1 + \triangle t$ and calculating an average ratio for every trajectory as proposed in [24].

The fractal dimension (FD), which we used as another feature, provides an index of complexity. The FD compares how detail in a pattern changes with the scale at which it is measured [68]. We used the definition of the FD from Sevcik [69]:

$$\text{FD} = 1 + \frac{\log (L)}{\log (2N - 2)}, \tag{15}$$

where $L$ is the contour length of the trajectory in the unit square and the number of points of the trajectory is given by $N$.

The radius of gyration of a trajectory (RG) is defined as

$$\text{RG} = \frac{1}{N} \sum_{i=1}^{N} |r_i - r_s|^2 = \frac{1}{N} \sum_{i=1}^{N} |r_i - \bar{r}|^2, \tag{16}$$

where $r_s = \bar{r}$ denotes the average position of the trajectory.

As a measure for the linearity of a trajectory the efficiency (E) is used. It is defined as

$$\text{E} = \frac{|x_{N-1} - x_0|^2}{(N - 1) \sum_{i=1}^{N-1} |x_i - x_{i-1}|^2}, \tag{17}$$

where $x_0$ is the start position of the trajectory. The efficiency relates the squared net displacement to the sum of the squared displacements.

Similarly, the straightness (S) can be used, which is defined as

$$\text{S} = \frac{|x_{N-1} - x_0|}{\sum_{i=1}^{N-1} |x_i - x_{i-1}|}, \tag{18}$$

and relates the net displacement to the sum of step lengths.

The last feature used is the Gaussianity (G). Proposed by Ernst *et al.* [70] the gaussianity checks the Gaussian statistics on increments within a trajectory. It is defined as

$$\text{G} = \frac{\langle r^4_n \rangle}{2 \langle r^2_n \rangle^2}, \tag{19}$$

with

$$\langle r^4_n \rangle = \frac{1}{N - n} \sum_{i=1}^{N-n} |x_{i+n} - x_i|^4.$$

All eight features are summarized and provided with literature references in Table 1.

**Table 1. Table of features used to characterize the trajectories of the various segregation mechanisms.**

| feature | references |
|---------|-----------|
| MSD | [20–22, 24, 25, 71] |
| MSDR | [22, 24, 71] |
| Alpha | [22, 24] |
| FD | [22–24, 68, 69] |
| RG | [23, 72, 73] |
| E | [22, 24] |
| S | [22, 24] |
| G | [22, 24, 70] |

Table of features used to characterize the trajectories of the various segregation mechanisms. The low-dimensional input vectors for the classifiers are constructed from these features.

## Results

With our MD framework, we were able to simulate eight different cell types. They are constructed via four different modes of segregation: as entropic repulsion of chromosomes is the consequence of basic physical principles, there's no sense in turning it off and it thus is present in all our simulations. However, we can run simulations in which either ParAB (=dParAB) or SMC (=dSMC) or even both proteins are not active (=dSMCdParAB) as in knock-out mutants. In our hypothetical wild type all segregation mechanisms are active. Together with the two different schemes for replication, the track and the factory model, we get for our classifiers a total of eight classes to discriminate. They are listed in Table 2.

For classification of the different segregation mechanisms we always used the trajectories of the newly duplicated *ori* which is pulled towards the cell pole by the ParAB system (if ParAB is active) as input for our classifiers. In total we simulated 24,000 trajectories that were evenly distributed among the cell types. Examples of such trajectories are shown in S2 Fig in S5 Appendix. In the trajectories one can see that the effect of ParAB is particularly pronounced. As a consequence, the cell types with activated ParAB typically show a directed movement of the *ori* towards the cell pole whereas cell types in which ParAB is not active show less directional movement. Furthermore, without the directionality in segregation provided by the ParAB system, the *ori* does not show a clear preference towards one of the two cell poles in its movement. It should also be noted that in the factory model replication takes place at the cell center and the *ori* migrates from there to one of the cell poles. This is not the case in the track model, so that the *ori* can travel longer distances in this model when it is pulled towards the more distant

**Table 2. Cell types analyzed in our simulations.**

| Track model | Factory model |
|-------------|---------------|
| WT | WT |
| dSMC | dSMC |
| dParAB | dParAB |
| dSMCdParAB | dSMCdParAB |

Cell types analyzed in our simulations. In wild type (WT) cells all three segregation mechanisms are present. Furthermore, we can build mutants in which one of the proteins is knocked out (dSMC, dParAB) or in which both segregation proteins are knocked out (dSMCdParAB).

cell pole. As a result, trajectories with activated ParAB in the factory model resemble those without ParAB in the track model in terms of the distance traveled by the *ori*. We also measured the degree of separation of the two chromosomes after termination of replication for all cell types. The results are summarized in S4 Table in S5 Appendix. The results demonstrate that the ParAB system not only has a strong influence on the movement of the *ori* but also leads to an increased effectiveness of overall chromosome segregation. It can be seen that cells lacking ParAB are still able to separate their chromosomes during the replication period, but not as reliable as they do with activated ParAB.

In the following, we evaluate the performance of our ML models in the classification of the different cell types. We start with the approach of high-dimensional input vectors.

## Classification based on high-dimensional input vectors

One of the advantages of the rescaling scheme proposed by Muñoz *et al.* is that trajectories of different spatiotemporal scales can be compared [21]. This can be seen in the plots of Fig 4 where the average distances traveled by the *ori* as obtained from the raw data are compared to the rescaled values.

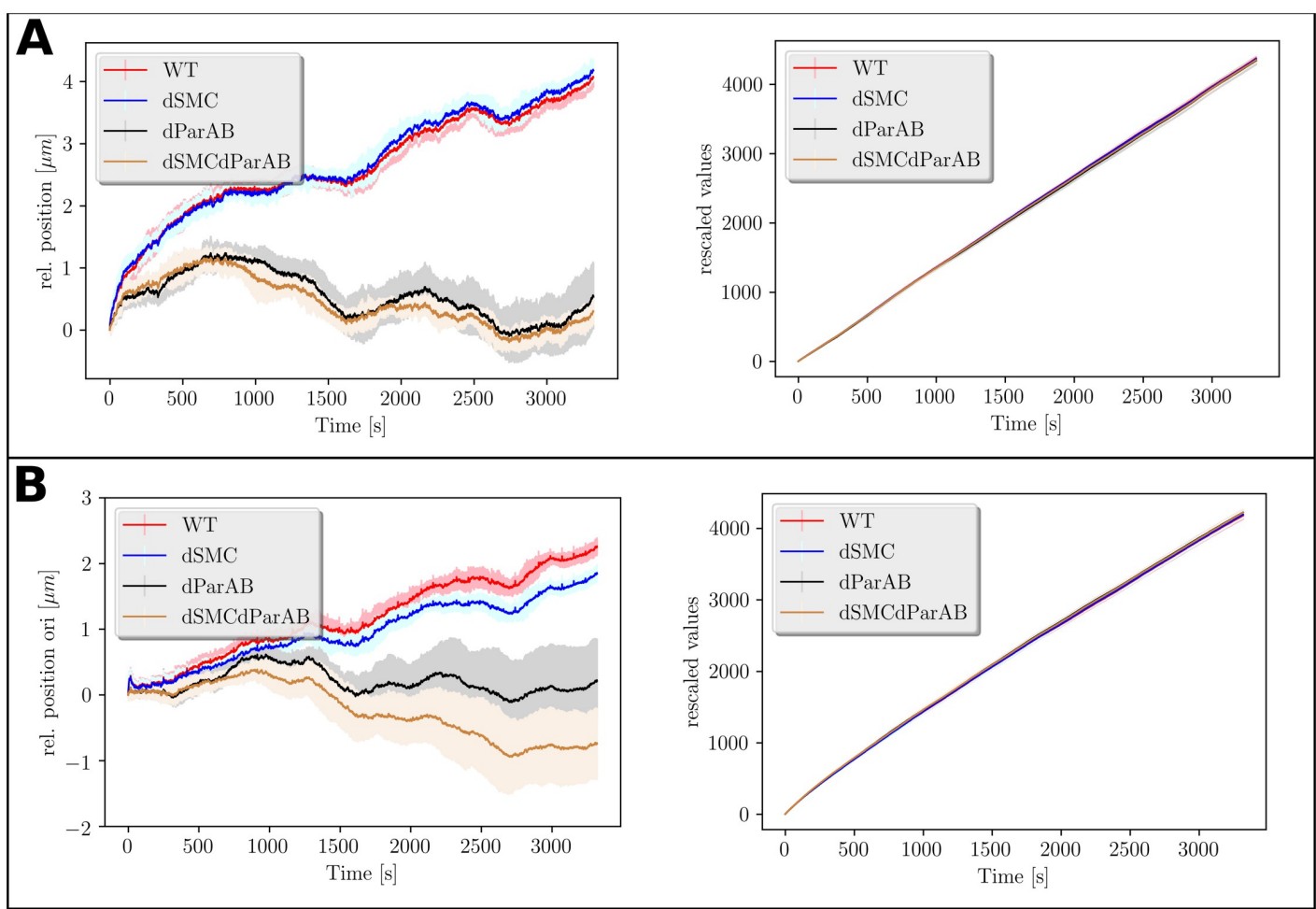

**Fig 4. Averaged trajectories for all cell types. A**: Average trajectories of *ori* for cell types with the track model of replication (left) and average trajectories after rescaling (right). The mean values were calculated over 3000 trajectories per class. The shaded areas show the standard deviations. **B**: Same plots as in A for the factory model of replication.

**Table 3. Overall prediction accuracies for classifiers using high-dimensional input vectors.**

| Model | Acurracy (train set) | Acurracy (test set) |
|---|---|---|
| Random forest | 0.994 | 0.915 |
| Gradient Boosting | 1.0 | 0.965 |
| Logistic regression | 0.950 | 0.932 |
| SVM | 0.949 | 0.925 |

Overall prediction accuracies of the classifiers on the train and test data using high-dimensional input vectors.

In the plots of Fig 4 it can be seen that rescaling sums up the individual trajectories to comparable magnitudes. At the same time, the rescaled trajectories make it impossible to distinguish between cell types with the naked eye. On the other hand, the mean values of the unscaled trajectories show a division into trajectories in which ParAB is active and those in which ParAB is not active. However, one should note that while the trajectories of the ParAB knock-out mutants can be well discriminated from those of cells with active ParAB at the end of segregation, it is much more difficult to do so at the start of the simulations and for shorter trajectories.

In order to test the classification performance of our classifiers they were trained with the training data and we evaluated the overall prediction accuracies on both the known training data and the unknown test data. For this, the number of correct predictions of a classifier was divided by the total number of predictions. In Table 3 the results of this evaluation are shown.

Table 3 shows very high prediction accuracies for the classifiers on both training and test data. The classifiers were able to learn the training data as well as to correctly assign the unfamiliar test data. However, the results for the RF classifier indicate a tendency to overfitting since we recognize a gap of 7.9% in accuracy between training and test data. This could be a hint that the classifier might have difficulties in generalizing from the training data. On the other hand, especially the linear classifiers show similar prediction accuracies on both training and test data.

To analyze the performance of the classifier in more detail, we have plotted the confusion matrices in Fig 5. This allows us to directly compare the predictions of the classifiers with the actual labels of the data. In this way, one can determine and visualize the number of correct and incorrect predictions provided by the classifiers for each class. Consequently, the strengths and weaknesses of the classifiers can be identified more precisely.

As can be seen from the confusion matrices in Fig 5, none of the classifiers confuses the two replication models. Instead, the dominant source of classification errors in the track model is the confusion of wild type cells with dSMC cells. In contrast, within the factory model, the most common mistake is made in discriminating dParAB and dSMCdParAB cells. Furthermore, the classifiers don't have difficulties in discriminating segregation with and without ParAB as we expected from the raw data. These observations from the confusion matrix can be quantified by calculating the precision and recall values. The precision value is defined as the fraction of correct predictions among all predictions. Thus, it quantifies how often a classifier is correct predicting a certain class. The recall value is defined as the fraction of correct predictions of a given class relative to the total number of members of this class. With this it tells us how often a specific class is predicted correctly. The precision and recall values calculated from the confusion matrix in Fig 5 are summarized in Table 4.

In general, we find very high precision and recall for all our classifiers. For a more detailed analysis of the precision and recall values, it makes sense to look at them separately for each

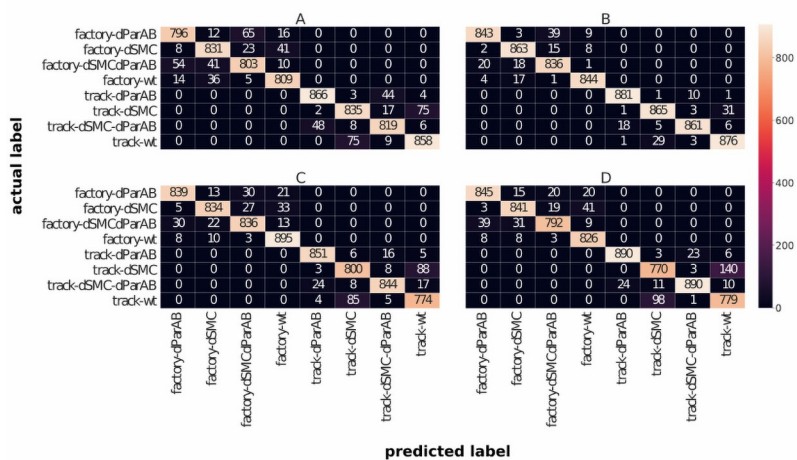

**Fig 5. Confusion matrices for all four classifiers using high-dimensional input vectors.** The predicted labels (horizontal axis) are compared with the actual labels of the test data (vertical axis). The numbers in the matrix entries indicate the number of trajectories from the test dataset that were assigned the respective label by the classifiers. **A**: Confusion matrix for RF classifier. **B**: Confusion matrix for GB classifier. **C**: Confusion matrix for LR classifier. **D**: Confusion matrix for SVM classifier.

replication model. For the track model of replication, we find highest precision and recall scores for cell types lacking ParAB. This matches the observation from the confusion matrices that for cells with the track model of replication, the most common error is the confusion of the cells with activated ParAB, namely `WT` cells and `dSMC` cells. These cell types are more similar for the track model than are the cell types without ParAB. If we consider cell types with the factory model of replication, we realize that the precision and recall values are highest for cells in which ParAB is active. Accordingly, the most common mistake made by classifiers here was confusing `dParAB` and `dSMCdParAB`. Consequently, the picture changes in the factory model, i.e. when replication takes place in the cell center and the *ori* has a shorter distance to travel to the cell pole. We will return to this observation in the discussion and discuss possible interpretations for it. Up to this point, we can conclude that the classifiers typically only mix up cells in which ParAB is either active or inactive. Within which of these groups the error rate is higher depends on the replication model.

**Table 4. Precision and recall values of the classifiers for high-dimensional input.**

| | Precision | | | | Recall | | | |
|---|---|---|---|---|---|---|---|---|
| Cell type | RF | GB | LR | SVM | RF | GB | LR | SVM |
| Track-WT | 0.910 | 0.958 | 0.876 | 0.833 | 0.911 | 0.964 | 0.892 | 0.887 |
| Track-dSMC | 0.907 | 0.961 | 0.890 | 0.873 | 0.899 | 0.961 | 0.890 | 0.843 |
| Track-dParAB | 0.950 | 0.978 | 0.965 | 0.974 | 0.944 | 0.987 | 0.969 | 0.966 |
| Track-dSMCdParAB | 0.921 | 0.982 | 0.967 | 0.971 | 0.930 | 0.967 | 0.945 | 0.952 |
| Factory-WT | 0.924 | 0.979 | 0.930 | 0.922 | 0.936 | 0.975 | 0.977 | 0.978 |
| Factory-dSMC | 0.903 | 0.958 | 0.949 | 0.940 | 0.920 | 0.972 | 0.928 | 0.930 |
| Factory-dParAB | 0.913 | 0.970 | 0.951 | 0.944 | 0.895 | 0.943 | 0.929 | 0.939 |
| Factory-dSMCdParAB | 0.896 | 0.938 | 0.933 | 0.950 | 0.884 | 0.955 | 0.928 | 0.909 |

Precision and recall values for all classifiers based on high-dimensional input vectors.

## Classification based on low-dimensional input vectors

After the first good success in classifying the different cell types based on high-dimensional input vectors, we used low-dimensional vectors as input for the classifiers in a second approach. Typically, one hopes for reduced computational effort and faster classifications from such a dimensional reduction. Moreover, overfitting of the classifiers can be prevented in many cases as a result of feature reduction as this reduces the chances of fitting aspects of the data that are not generalizable outside the data [20]. We constructed our input vectors with the statistical features of the trajectories descrived above, i.e. the mean squared displacement (MSD), the exponent alpha (Alpha), the mean squared displacement ratio (MSDR), the fractal dimension (FD), the radius of gyration (RG), the efficiency (E), the Gaussianity (G) and the straightness (S). Again, we started our analyses by measuring the overall prediction accuracies of our classifiers on both training and test data. The results are shown in Table 5.

Analysis of the results from Table 5 provides several interesting observations. For the tree-based classifiers we observe an increased overall prediction accuracy compared with the values found for the approach with high-dimensional input vectors. Furthermore, we observe that the RF now shows a prediction accuracy of 97.4% on the training data and 96% on the test data. Thus, the gap between prediction accuracy on training and test data drops to 1.4% while it was 7.9% in the case of high-dimensional input vectors. Also for the GB we do not find much difference in terms of accuracy on the training data and the test data. In this respect, it can be stated that the tree-based models have both improved their accuracy and exhibit fewer effects of overfitting by using low-dimensional input vectors. However, we find a worse performance of the linear classifiers with the low-dimensional input vectors compared to the high-dimensional ones since both linear classifiers, LR and SVM show a clear drop in their overall prediction accuracy. We further investigate these results in the confusion matrices shown in Fig 6.

The plot of the confusion matrix in Fig 6 again shows a grouping of the errors made depending on the replication model considered. Looking first at the tree-based classifiers (subplots A and B) we note that for the factory model of replication classification errors only arise due to a confusion of dParAB with dSMCdParAB. Within the track model of replication there are still some further classification errors in which WT is wrongly labeled as dSMC or the other way around. Interestingly, the linear classifiers (subplots C and D) also don't show any misclassifications for cells with activated ParAB in the factory model of replication. Instead, there are many more errors due to confusion of cell types lacking ParAB, which explains the generally worsened overall prediction accuracy. We can extend these results by looking at the precision and recall values shown in Table 6.

The results of the precision and recall values from Table 6 confirm the observation that the tree-based classifiers show increased classification strengths using low-dimensional input vectors while the performance of the linear models becomes worse. Interestingly, we can also

**Table 5. Overall prediction accuracies for classifiers using low-dimensional input vectors.**

| Model | Acurracy (train set) | Acurracy (test set) |
|---|---|---|
| Random forest | 0.974 | 0.960 |
| Gradient Boosting | 1.0 | 0.973 |
| Logistic regression | 0.889 | 0.860 |
| SVM | 0.891 | 0.872 |

Overall prediction accuracies of the classifiers on the train and test data using low-dimensional input vectors.

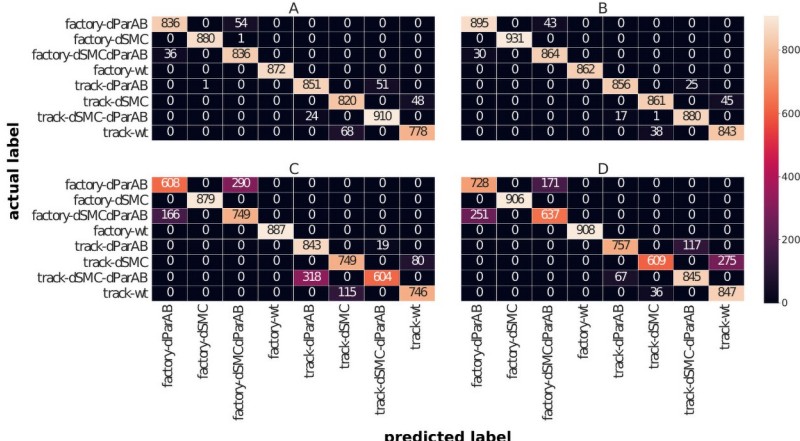

**Fig 6. Confusion matrices for low-dimensional input vectors.** Confusion matrices for the classifiers using statistical features of the trajectories to construct low-dimensional input vectors. The predicted labels (horizontal axis) are compared with the actual labels (vertical axis). **A**: Confusion matrix for RF classifier. **B**: Confusion matrix for GB classifier. **C**: Confusion matrix for LR classifier. **D**: Confusion matrix for SVM classifier.

confirm the previously made observation that the frequency of errors, which are either a confusion of cell types with activated ParAB or without the action of ParAB, is dependent on the replication model. In the present case of low-dimensional input vectors, this is particularly striking due to the perfect prediction accuracies of all classifiers for WT cells and dSMC cells in the factory model. This shows that in the factory model, classification errors only occur for cell types in which ParAB is inactive, whereas in the track model, cell types with active ParAB are misclassified more frequently.

In summary, we can conclude that the linear classifiers become significantly worse when using statistical features as input. On the other hand, the tree-based classifiers increase their accuracy in classification and reduce overfitting. In the next section, we try to shed more light on these findings by analyzing which features are rated as most important by the classifiers.

**Feature importance.** One advantage of using low-dimensional input vectors is that the importance of the features for classification can be quantified. This opens new possibilities for making the classifiers computationally less demanding. For example, if one is able to identify features with high importance one can conclude that these are the drivers of the model

**Table 6. Precision and recall values of the classifiers for low-dimensional input.**

|  | Precision | | | | Recall | | | |
|---|---|---|---|---|---|---|---|---|
| Cell type | RF | GB | LR | SVM | RF | GB | LR | SVM |
| Track-WT | 0.942 | 0.949 | 0.903 | 0.755 | 0.920 | 0.957 | 0.866 | 0.959 |
| Track-dSMC | 0.923 | 0.957 | 0.867 | 0.944 | 0.945 | 0.950 | 0.903 | 0.689 |
| Track-dParAB | 0.973 | 0.981 | 0.726 | 0.919 | 0.942 | 0.972 | 0.978 | 0.866 |
| Track-dSMCdParAB | 0.947 | 0.972 | 0.970 | 0.878 | 0.974 | 0.980 | 0.655 | 0.927 |
| Factory-WT | 1.00 | 1.00 | 1.00 | 1.00 | 1.00 | 1.00 | 1.00 | 1.00 |
| Factory-dSMC | 0.999 | 1.00 | 1.00 | 1.00 | 0.999 | 1.00 | 1.00 | 1.00 |
| Factory-dParAB | 0.959 | 0.968 | 0.786 | 0.744 | 0.939 | 0.954 | 0.677 | 0.810 |
| Factory-dSMCdParAB | 0.938 | 0.953 | 0.721 | 0.788 | 0.959 | 0.966 | 0.819 | 0.717 |

Precision and recall values for all classifiers based on low-dimensional input vectors.

outcome. Subsequently, the least important features often might be omitted [20] simplifying the fitting procedure and making prediction faster. Furthermore, feature importance analysis might provide new insights into the segregation mechanisms by connecting them to quantitative features of the trajectories. In tree-based models there exist several ways to calculate the importance of the single features for the outcome of a classification. Here we use the built-in function `feature_importances_` of `scikit-learn` which is explained in [59]. In this approach, one calculates the Gini impurities before and after each split on a given feature and the total decrease in the node impurity related to the respective feature. Finally, the outcome is averaged over all trees in the ensemble [20]. For the calculation of feature importance in linear models we use the coefficients of the features in the decision function. Thereby one makes the assumption that a higher coefficient of a given feature indicates higher importance of this feature for the classification. This assumption is only valid for a SVM if one uses a linear kernel (as we did) to ensure that the hyperplane selecting the datapoints in the SVM is in the same space as the input features so that the coefficients can be compared. If this were not the case, the data transformed by the non-linear kernel function would be in another space which is not directly connected to the input space of our features and there would be no way to extract feature importance values from the coefficients. Furthermore, to get percentage values for the feature importance the input features had to be rescaled using the `StandardScalar` function implemented in `scikit-learn`. With this, the features were scaled to unit variance which makes the magnitudes of the coefficients comparable.

The results of our feature importance analyses are summarized in the plots of Fig 7.

In subplots A and B of Fig 7 the feature importance values for the tree-based and the linear classifiers are shown. As can be seen, all classifiers identify similar features as the most important ones. The top four of the most important features are Alpha, RG, FD and MSD. Furthermore, the MSDR is not considered to be very important especially for the linear classifiers. However, the differences between the importance assigned to the features are not very large. It is striking that the SVM considers $\alpha$ and FD to be by far the most important features while the other classifiers include all features in their decision with importances of more or less the same

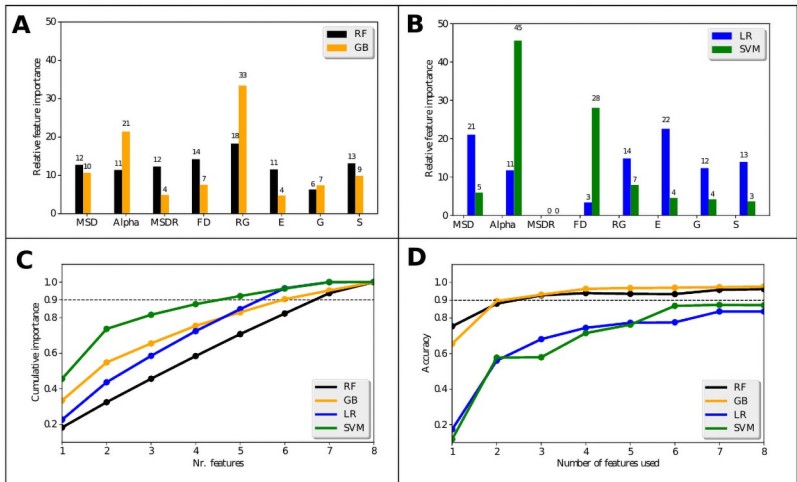

**Fig 7. Feature importance plots. A**: Bar chart of the relative feature importance for the tree-based classifiers. The importance values are given as percentages and are shown as numbers above the bars for a better overview. **B**: Bar chart of the relative feature importance for the linear classifiers. **C**: The plot shows the number of features required to reach a defined level of cumulative importance. The level of 90% cumulative importance is indicated by the dashed line. **D**: Overall prediction accuracy of the classifiers shown as a function of the number of features used for training.

order of magnitude. Thus, they use more of the available information by including all features in the decision. This is also illustrated by Fig 7C. Here, the cumulative importance is plotted as a function of the features used. It can be seen that the SVM already reaches 90% cumulative feature importance by the combination of the four most important features. This results in a steeper curve in the plot compared to the other classifiers which, on the other hand, show almost linear curves indicating that classification is more evenly distributed among the features. In Fig 7D we show the achieved accuracies of our classifiers if we only use a subset of the features as input. To do so, we successively removed the least important features from the input and then tested the accuracy of the classifier based on the remaining features. The graph shows that the tree-based classifiers already reach an accuracy of 90% with the two most important features. The linear classifiers need more features (5 to 6), to reach an accuracy of ∼80%. One could use this graph as a tool for feature selection, i.e. define a threshold of accuracy to be reached and omit the remaining features which are not needed. In this case the tree-based models would require fewer features while still achieving a better accuracy than the linear classifiers. To better understand the results of feature importance analysis, we need to ask ourselves what each feature tells us about a trajectory. We will discuss this further in the discussion below.

## Classification of short trajectories

In real experiments it is likely that one cannot provide high-resolution time-lapse data over 50–60 minutes. Therefore, it is interesting to check if the classifiers are also capable of discriminating segregation mechanisms of shorter trajectories. It was demonstrated, that a random forest algorithm is capable of characterizing very short trajectories by applying the normalization protocol of [21] for simulated diffusion trajectories. Thus, we again used this approach with high-dimensional input vectors for our segregation trajectories and cut the trajectories into smaller segments down to a length of 5s (= 5 data points). Thereupon we again trained our four classifiers on the data and evaluated the accuracies on the unseen test data. The results of the overall classification accuracies as a function of the length of the trajectories are shown in Fig 8.

The results of Fig 8 demonstrate, that our classifiers, are also capable of classifying very short trajectories in agreement with the finding of [21]. Additionally, we find that for extremely short trajectories with lengths below 10s the tree-based classifiers show better classification results than the linear classifiers. Both tree-based classifiers reach accuracies of more than (80%) on trajectories below 50 s length. This is especially important since it seems impossible to differentiate trajectories of such a small length with the human eye and thus the machine learning algorithm provides a very useful tool, here. All classifiers reach scores in the range of their overall prediction accuracies for trajectories of full length after ∼ 500$s$. This indicates that trajectories of this length already are sufficient to obtain good classification result.

Another challenge that might arise within the classification of experimental data is that the temporal resolution at which the experimental trajectories are measured may vary. As long as we have data with the same temporal resolution, we can simply optimize and train a classifier on this temporal resolution. However, one may end up with a mixed data set containing trajectories of different temporal resolution.

To investigate this case, we varied our trajectories with an original temporal resolution of 1s to create a corresponding data set of different temporal resolutions. To do this, one of the following temporal resolutions was randomly selected for each of the trajectories: {1s, 2s, 3s, 4s, 5s, 10s, 15s, 20s, 30s, 40s, 50s, 100s}. To create this resolution from the original trajectory,

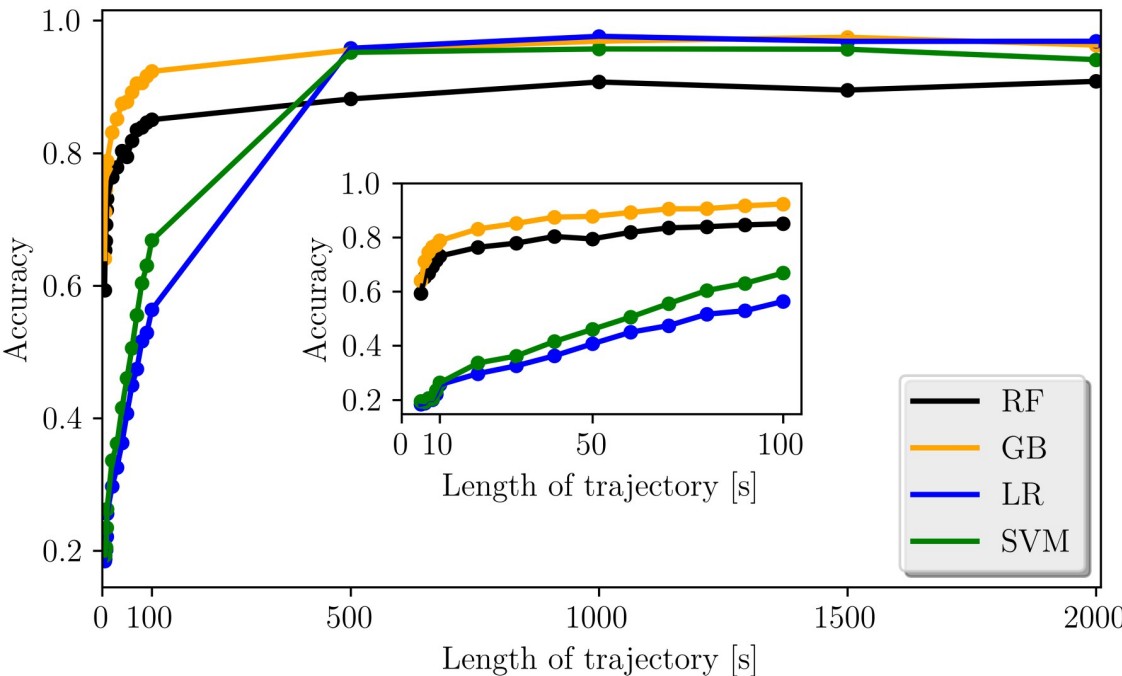

**Fig 8. Accuracy of the classifiers on short trajectories.** Overall prediction accuracies of the four classifiers as a function of the length of the trajectories. The subplot provides an enlarged view of the very first part of the graph with the length of the trajectories ranging from 5s to 100s.

every $n$-th point was selected from the original trajectory to realize a temporal resolution of $n$ secs. As a result, trajectories of different temporal resolution are obtained, which, however, also consist of different numbers of measurement points. To be able to classify these trajectories, the measurement points of the trajectories were interpolated in the following to obtain input vectors of the same dimension for the classifiers. This procedure is illustrated by an example in S3 Fig in S5 Appendix. In the following, we again split the new set of trajectories of different temporal resolutions into a training dataset (70%) and a test dataset (30%). In the following, the overall prediction accuracies of the classifiers were evaluated for the approach of high-dimensional input vectors as well as for the approach of low-dimensional input vectors. The values are listed in Table 7.

The results from Table 7 show that the overall prediction accuracies of all classifiers are lower for the dataset of mixed temporal resolutions compared to the previous results, as one

**Table 7. Overall prediction accuracies for trajectories of changing temporal resolutions.**

| | **High-dimensional** | | **Low-dimensional** | |
|---|---|---|---|---|
| **Model** | **% train set** | **% test set** | **% train set** | **% test set** |
| Random forest | 0.988 | 0.720 | 0.946 | 0.917 |
| Gradient Boosting | 1.0 | 0.812 | 1.0 | 0.932 |
| Logistic regression | 0.763 | 0.697 | 0.663 | 0.652 |
| SVM | 0.749 | 0.702 | 0.795 | 0.783 |

Overall prediction accuracies of all classifiers applied to the data set of trajectories with different temporal resolutions. The results of the classifiers are shown for both cases using either high-dimensional or low-dimensional input vectors. The prediction accuracies are shown for evaluation on the training data and on the test data.

would expect. Furthermore, we notice that the classification based on the high-dimensional input vectors is much less accurate than the approach with the low-dimensional input vectors. In addition, the two tree-based classifiers show significantly better results than the linear models. In fact, with the tree-based classifiers using the low-dimensional input vectors, we still achieve classification accuracies of over 90% on the test data. Thus, we conclude that with the use of tree-based classifiers, a good discrimination of segregation trajectories is possible even for the case of different temporal resolutions of the trajectories.

## Discussion

In the present work, we were able to extend our existing MD framework from [4] to combine different replication and segregation models to eight different cell types for which we could simulate trajectories of *ori*. Subsequently we trained ML models based on these synthetic trajectories in this way providing a proof of principle for the classification of different segregation mechanisms. Furthermore, we compared different approaches of feature design for the ML models and also challenged our classifiers with truncated trajectories or data of different temporal resolution. In the following, we will first discuss the findings from the implementation of different segregation mechanisms in our MD framework. We then review the results of the classifications using the different feature designs as well as the performance of our classifiers for truncated trajectories.

### MD implementation of segregation mechanisms

We successfully implemented three of the most prominent segregation mechanisms within our MD framework, namely the entropic segregation of chromosomes, the partitioning protein system ParAB and the action of SMC proteins. Entropic segregation of chromosomes relies on the mechanical properties of the chromosome and its confinement into the cell's interior. Thus, it is a purely physical mechanism and should contribute to chromosome segregation in all bacteria [12, 13, 15, 33]. In a previous study, we have already shown that a model of entropic segregation is able to reproduce experimental data of chromosome segregation in *B. subtilis* [4]. Therefore, we implemented entropic repulsion of chromosomes as a fundamental component in all our simulations. The ParAB system is the closest analog to the eukaryotic mitotic apparatus [8]. It is especially associated with the directed movement of the origins to the cell poles [43]. In most bacteria like *C. crescentus*, the ParAB system is crucial for the segregation of the origins to the cell poles. However, *B. subtilis* mutants lacking ParA were shown to segregate their chromosomes as assayed by the production of anucleate cells [8, 42]. We can therefore assume that ParAB has a crucial influence on the segregation of chromosomes and in particular on the movement of the *oris* to the cell poles, but it may also be possible to segregate chromosomes without the help of ParAB. Thus, while in our hypothetical wild type version of segregation, all three segregation mechanisms work simultaneously together, we also tested mutants in which ParAB is knocked out. We did the same with the second important protein, SMC. It was shown that SMC condensin complexes also play crucial roles in the organization of chromosomes and assist the ParAB system in the segregation of origins through juxtaposing the chromosomes [8, 10, 48]. Thus, we have created a wild type version (`WT`) in which all the segregation mechanisms are active, accompanied by two knock out mutants, in which either ParAB (`dParAB`) or SMC (`dSMC`) are disabled, and a double knock out mutant (`dSMCdParAB`), where both ParAB and SMC are inactive. Besides this, we used two different models for replication, the track model and the factory model which differ in the localization of the replication process. Typical model organisms associated with the two replication schemes are *E. coli* (track model) and *B. subtilis* (factory model) [1, 30, 31].

The results of our simulations indicate that the ParAB-assisted segregation is the most effi-cient mechanism in our model. In both replication models the `WT` and the `dSMC` cells showed a far more directed movement of the *ori* as a result of the pulling force of ParAB. Furthermore, the overall segregation of the chromosomes is more effecient in the presence of ParAB. Here, separation of the chromosomes is almost completely achieved within replication, while cells lacking ParAB only reach separations between 40%–65% (see S4 Table in S5 Appendix). At this point, however, it should be noted that our implementation of the ParAB system is based on a simplified assumption. Thus, although we were able to accurately derive the pulling force of the ParAB system in our simulations as proposed by the DNA-relay model [6], we did not consider the spatial distribution of ParA in the cell due to lack of corresponding data. In this respect, our constant force acting through the ParAB system on the *ori* represents an overesti-mation of the actual translocation effect by ParAB, since this force actually occurs only when the ParB/*parS* partition complex interacts with ParA. However, since the focus of our study was to provide a proof of principle for the classification of segregation patterns, we did not elaborate further on this point.

The action of the SMC proteins compact the spatial extension of the chromosome. Espe-cially the results of our simulations with the factory model indicate that deletion of SMC results in a drop of the distance traveled by *ori* compared to the wild type. In general, we con-clude that in both replication schemes the wild type shows the most efficient segregation of *ori* to the opposite cell pole. This might be a hint why bacteria developed such a variety of segrega-tion mechanisms. Thereby, they are enabled to survive the failure of one mechanism if others are provided to ensure faithful separation of the genetic material. For example, there are many bacteria which are capable of segregating their chromosomes almost normally while lacking SMC [74] and even the absence of ParAB is not lethal in every case [8, 42]. In this context, it seems as if entropic separation as a purely physical mechanism provides the basis of the segre-gation in bacteria since it cannot be eliminated. It is also tempting to speculate that for early live the existence of a separation mechanism which does not need the costly development of a complex machinery seems to be a valuable opportunity. In the further course of evolution, dif-ferent mechanisms could then have developed in parallel, which contribute to a diversification of the segregation of the genetic material. Of course, this presents us with the challenge to dis-criminate the various segregation mechanisms of a species by, e.g., using experimental data from single-particle experiments. Against this background, the automatic classification of seg-regation trajectories by ML models as discussed in the next section is an exciting possibility.

## Classification with high-dimensional input vectors

In our study, we constructed eight different cell types from different replication and segrega-tion mechanisms. To classify them with our ML models, we have rescaled in a first approach the trajectories following the procedure of Muñoz *et al.* [21] to construct high-dimensional input vectors. We found that all our four classifiers have accuracy scores above 90%. Thus, we can conclude that the proposed model from [21] does not only yield excellent results for the classification of diffusion models but also for our simulations of bacterial chromosome segre-gation. Furthermore, our work shows that the corresponding feature design also leads to good results for the additional classifiers we use (GB, LR and SVM). Interestingly, our results suggest that the linear models, even if they showed a lower accuracy on the test data, showed a lower tendency to overfitting compared with the tree-based model. A first result from the confusion matrices was that none of the classifiers showed problems discriminating the track model and the factory model of replication. Obviously, the localization of replication within the cell has a prominent influence on the trajectories. Since the *ori* in the factory model only has to segregate

from the cell center to the cell pole, whereas the distance to be covered in the track model can be significantly longer depending on the starting point of replication, it seems understandable that this is reliably detected by the classifiers. However, this is not visible in the normalized trajectories. Therefore, the machine learning algorithms show remarkable classification strength here. In-depth analysis of the confusion matrices showed further that the classifiers mostly made errors by confusing cells with ParAB (`WT`, `dSMC`) with cells without ParAB (`dParAB`, `dSMCdParAB`). From this we conclude that the action of ParAB is very prominent in the simulations. These results are consistent with observations in the literature that ParAB is one of the most important segregation proteins with direct influence on the movement of *ori* [1, 6, 8].

Another interesting result is the fact that the frequency with which the classifiers mix up either cells in which ParAB is activated or those in which ParAB is not activated depends on the replication model considered: For the track model of replication cell types in which ParAB is active are more often confused by the classifiers than those in which ParAB has been deactivated. This suggests that for the track model of replication, the trajectories of cells in which ParAB is active are more similar than those of cells in which ParAB is inactive. In contrast, within the factory model of replication the opposite is true. Here, the cell types in which ParAB is not active are more frequently confused with each other than those in which ParAB is active. From this we conclude that the effect of ParAB is particularly dominant in the track model of replication, in which the *ori* was transported over a longer distance and cells in which ParAB is active are therefore particularly similar. In the factory model of replication, however, the two cell types in which ParAB is active are less likely to be confused by the classifiers. This could be an indication for the fact that in the factory model, the deactivation of SMC is more important because replication takes place in the middle of the cell, where it could be particularly important that SMC topologically divides the separating daughter chromosomes.

Furthermore, we found that in general the classifiers had lowest precision in identifying the `WT` cells. This is explainable by the fact that here all segregation mechanisms are incorporated and thus this class has closest resemblance to all other classes.

Taken together we can state that we successfully applied the normalization protocol from [21] to our chromosome segregation simulations. We furthermore demonstrated that the high-dimensional input vectors can also be used as input for other algorithms besides the random forest classifier whereby the linear models showed a good tradeoff between accuracy and avoidance of overfitting. However, the highest prediction accuracy on both test and training data is achieved with the GB classifier. An important advantage of the classification with normalized trajectories is that the spatio-temporal scale from which the data is obtained is not relevant. Thus, one could, for example, compare segregation trajectories from bacterial species of different sizes and with different lengths of the replication cycle. In future applications it would be interesting to include several experimental trajectories into our datasets to stimulate further generalization of the models towards experimental data and also to test the application on real data.

### Classification with low-dimensional input vectors

In many machine learning problems it is desirable to reduce the dimensionality of the input space for a classifier. To achieve this, we used a set of eight statistical features which we calculated for every trajectory and used as a new low-dimensional input vector for our classification routines. The analysis of the resulting accuracies on the test and train data revealed that the reduced dimensionality of the input both enhanced the accuracies of our tree-based classifiers and reduced the amount of overfitting. On the other hand, the linear classifiers showed a significant drop in their classification accuracy. Consequently, we assume that the linear

classifiers are a good choice for classification using the high-dimensional input vectors as explained above while the tree-based classifiers work better with the statistical features as low-dimensional input vectors. Overall, also for the case of low-dimensional input features, it turns out that the best prediction accuracy is achieved with the GB classifier. The iterative approach of the GB classifier to gradually train new decision trees based on the previously made errors of the ensemble thus turns out to be the most successful algorithm for classifying the segregation mechanisms in our study. In addition to the overall accuracy values of the classifiers, we found no significant differences in the errors made by the classifiers as shown in the confusion matrices. However, using the feature-based approach has the advantage that we can compute the relative importance of the features for the classification result.

Here, the most important features were the exponent alpha, the radius of gyration, the fractal dimension and the mean-squared displacement. The mean-squared displacement ratio showed particularly low importance values.

As a result we found that the tree-based classifiers already perform very well in the classification task when using only the two or three most important features as input data. Also, removing the mean-squared displacement ratio from the input vectors did not dramatically reduce the accuracy of the classifiers. The fact that variables such as the exponent alpha and the fractal dimension are classified as particularly important features can be understood from their physical interpretation. Typically, the exponent alpha is used to characterize different types of diffusion. For values of $\alpha > 1$ one has directed motion while $\alpha < 1$ characterizes anomalous diffusion and normal diffusion has values of $\alpha \approx 1$ [22, 24]. In our simulations, the effect of ParAB provides a more pronounced directionality of the motion of the *ori* and thus trajectories in which ParAB is active more closely resemble directional motion than those in which ParAB is inactive. In the same way, the fractal dimension takes values around 1 in the case of straight trajectories while random trajectories lead to values of the fractal dimension around 2 [68]. Thus, this feature helps the classifiers to distinguish trajectories with a rather directional motion (WT, dSMC) from those that are more like a diffusive motion. We also found that the linear classifiers performed less well than the tree-based classifiers when using the low-dimensional input vectors. One reason for this could be that some of the selected features correlate with each other. This is particularly evident in the case of efficiency, which relates the square net displacement to the sum of the squared displacements and the straightness, which relates the net displacement to the sum of the step lengths [22, 24]. Such a correlation of different features causes problems for the linear classifiers which add the individual effects of all features. Consequently, it is no longer possible to determine to which of two correlating features a specific effect belongs. It becomes clear at this point why the low-dimensional approach works better with the tree-based classifiers. In this case it was found that it makes sense to use all features as input for the classifiers, since the importance values for all features were of comparable size and thus all available information can be used for an optimal prediction accuracy.

## Classification of short trajectories

As a final test for our classifiers we presented them trajectories consisting of fewer datapoints. In this way, we wanted to test if classification of segregation mechanisms is also possible for trajectories of several seconds, as they might be available from experiments. Muñoz et al. were able to characterize trajectories of their diffusion models consisting of only 10 points using their normalization protocol [21]. Thus, we decided to present our classifiers equally short normalized trajectories as high-dimensional input vectors. We found that all four classifiers are capable of reaching very good classification accuracies for the shortened trajectories.

However, the gradient boosting classifier performed best at this task, being able to classify trajectories of only 5 datapoints with an accuracy of $\approx 80\%$. Therefore, it looks promising that with our approach it will be possible to also classify short experimental data series.

Finally, we examined the case of a data set with trajectories of different temporal resolutions. In this case, the classification accuracy of all classifiers was lower than for the case of trajectories with a uniform temporal resolution, as expected. However, we found that the tree-based classifiers still were able to reach prediction accuracies of more than 90% on the test data when using the statistical features as low-dimensional input vectors. This was not possible with the linear classifiers.

## Outlook

The results of this work show that it is possible to analyze different replication and segregation mechanisms of bacterial chromosomes with MD simulations and to classify them by ML models. The proof of principle provided in this work motivates further development of the models in future studies. On the one hand, a more detailed implementation of the segregation mechanisms of the ParAB system and SMC in the MD framework is possible. For example, the inclusion of the spatial concentration of ParA in the cell could have an effect on the pulling force exerted by the ParAB system at the *ori*. Moreover, the mobility of SMC complex along the chromosome [75, 76] could be implemented by varying the potentials in the MD framework. In addition, studies on different model organisms show that the effects of ParAB and SMC can vary from species to species [4, 5, 9, 41, 42]. Therefore, it might be interesting to simulate corresponding cell types in which the proteins contribute differently to the segregation of the chromosomes. This would then allow a more accurate classification of as yet unknown species along known segregation patterns.

Another exciting next step would be to incorporate experimentally measured trajectories into the classification. Since it is difficult to provide sufficient data for training ML models, a mixed data set of synthetic MD trajectories and trajectories actually measured in experiments can be used.

Furthermore, the results of this work illustrate the potential benefit of applying ML models to trajectories from single particle tracking experiments. For example, one could also strive to classify the movement of replisomes in the cell to distinguish whether their movement bares more resemblance to the track model of replication or the factory model [15, 29, 32]. Corresponding synthetic trajectories could also be produced for this purpose with the presented MD framework.

## Conclusion

We developed a MD framework to simulate various segregation and replication mechanism of bacterial DNA. Using this we produced a large number of trajectories that we used as input for four machine learning algorithms for classification of the segregation mechanisms. We trained the machine learning models with both high-dimensional input vectors constructed from the complete rescaled trajectories and with low-dimensional input vectors built from a set of statistical features. We found high prediction accuracies of the classifiers in both cases. Furthermore, our results suggest to use the linear classifiers for the high-dimensional input vectors as they show less overfitting than the tree-based classifiers. The tree-based classifiers should be used on the low-dimensional input vectors where they reduced overfitting while yielding excellent accuracy scores. The advantage of using machine learning algorithms here is that it enables automated analyses of a large amount of data within seconds. For further studies, it

would be important to test the classifiers on experimental data. This would require a sufficiently large data set of well-defined experimental segregation trajectories.

## Supporting information

**S1 Appendix. Model for DNA** [1, 3, 4, 14, 26, 33, 36–38, 47, 77–85].
(PDF)

**S2 Appendix. Simulation setup** [4, 14].
(PDF)

**S3 Appendix. Acceleration of MD time** [4, 49, 51, 86–88].
(PDF)

**S4 Appendix. Nested cross-validation** [55, 58, 61].
(PDF)

**S5 Appendix. Hyperparameter tuning** [20, 57, 61].
(PDF)

## Acknowledgments

We thank Peter Graumann and Kurt Drescher for fruitful discussions on the topic and providing ideas for further developments of the models.

P.L. and D.G. conceived of the project. D.G. performed the simulations. P.L and D.G. wrote the paper.

## Author Contributions

**Conceptualization:** David Geisel, Peter Lenz.

**Data curation:** David Geisel.

**Formal analysis:** David Geisel.

**Funding acquisition:** Peter Lenz.

**Investigation:** David Geisel.

**Methodology:** David Geisel.

**Project administration:** David Geisel, Peter Lenz.

**Software:** David Geisel.

**Supervision:** Peter Lenz.

**Validation:** David Geisel.

**Visualization:** David Geisel.

**Writing – original draft:** David Geisel.

**Writing – review & editing:** David Geisel, Peter Lenz.

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
