## [Decision Letter · Decision Letter 0]

11 Oct 2021

PONE-D-21-29250Machine Learning Classification of Trajectories from Molecular Dynamics Simulations of Chromosome SegregationPLOS ONE

Dear Dr. Lenz,

Thank you for submitting your manuscript to PLOS ONE. After careful consideration, we feel that it has merit but does not fully meet PLOS ONE’s publication criteria as it currently stands. Therefore, we invite you to submit a revised version of the manuscript that addresses all the points raised during the review process.

We look forward to receiving your revised manuscript.

Kind regards,

Hans A. Kestler

Academic Editor

PLOS ONE

Journal Requirements:

3.Thank you for stating the following financial disclosure: 

"This work was supported by the Deutsche Forschungsgemeinschaft (DFG, TRR174)."

"We thank Peter Graumann and Kurt Drescher for fruitful discussions on the topic and providing ideas for further developments of the models.This work was supported by the Deutsche Forschungsgemeinschaft (DFG, TRR174).P.L. and D.G. conceived of the project. D.G. performed the simulations. P.L andD.G. wrote the paper.We declare no competing interest"

"This work was supported by the Deutsche Forschungsgemeinschaft (DFG, TRR174)."

Reviewers' comments:

Reviewer's Responses to Questions

**Comments to the Author**

1. Is the manuscript technically sound, and do the data support the conclusions?

Reviewer #1: Yes

Reviewer #2: Yes

2. Has the statistical analysis been performed appropriately and rigorously? 

Reviewer #1: Yes

Reviewer #2: Yes

3. Have the authors made all data underlying the findings in their manuscript fully available?

Reviewer #1: Yes

Reviewer #2: Yes

4. Is the manuscript presented in an intelligible fashion and written in standard English?

Reviewer #1: Yes

Reviewer #2: Yes

5. Review Comments to the Author

Reviewer #1: This manuscript is generally well written and argued. The scientific context of this research work is well introduced within the first chapters, while a solid description of the methods and results obtained are provided in the second part. According to the publication criteria of PLOS ONE, I consider this work to be potentially suitable for acceptance, following minor revision.

Please find below my point to point list of comments:

1. Typos:

. Line 61: "... we briefly summarise this replication...".

. Eq. 14: space after "with".

. Line 482: "... modes of segregation: As entropic..." not capital A

2. Line 271: "the mother chromosome was connected to the replication factory by additional harmonic springs". -- How? Through every bead in the chromosome or only a subset of those?

3. Line 292: "... two chromosome arms are connected by an additional SMC bond". -- How is this bond modeled?

4. Line 298: "... we followed the procedure from Lim et al. to get an estimate of the elastic force resulting from the dynamics of the singular loci of the chromosome". -- A quick recall of Lim et al. findings, namely that the origin of the segregation strength provided by the ParAB complex is given directly by the intrinsic elastic properties of the chromosome itself, may be useful to the reader for a complete understanding of the method within this work.

5. Line 438: "... we created low-dimensional input vectors from a set of statistical features... ". -- Here, did the authors use the original trajectories or the rescaled ones to extract the statistical features?

6. Line 447, MSD evaluation: what n (not capital) have been used for the evaluation of the MSD parameter from every trajectory, namely for the first statistical feature introduced? (Assumed obviously that this parameter needs then to be varied to evaluate some of the other features).

7. Line 450: The text describing the role of n in the formula is not completely clear to me in its formulation. I would describe it a bit more, maybe mentioning something like:"the index n indicates the length of the time-step considered in the evaluation of the MSD along the trajectory. For example, for n=2, the formula evaluates the MSD of the trajectory between every point and its evolution after 2 time steps".

8. Line 690: Though Fig. 8 is supposed to show some of the most relevant results of this work, I have not been able to find it within the provided pdf file. Is it possible that the mentioned figure is missing?

9. Line 773: "The results of our simulations indicate that the ParAB-assisted segregation is the most efficient mechanism. -- This have been proved to be true only within your model (therefore I would add "in our model" at the end of the sentence). As you mentioned later, its efficiency could be "an overestimation of the actual translocation effect by ParAB" due to the "implementation of the ParAB system being based on a simplified assumption".

Reviewer #2: In the article „Machine Learning Classification of Trajectories from Molecular Dynamics Simulations of Chromosome Segregation“, Geisel et al. expand on their previous work, presenting various models for the chromosome segregation process in bacteria. The article presents a novel and interesting approach of using results from molecular dynamics simulations to provide the large amount of data required to train a classifier. The simulations include four different segregation mechanisms, each simulated for a ‚track‘ and ‚factory‘ replication model which distinguish whether or not the replisomes are assumed to be located at a fixed position in the center of the cell.

The setup of the model is well-motivated and explained in great detail, as are all steps of the simulation procedure. All assumptions are shown to be in accordance with the literature. The authors also verified that this is the case for the forces acting in their simulation. It is also nicely shown that it is possible to use only a small number of statistical measures as inputs for the classifier instead of high-dimensional vectors, thus reducing the computational demand.

The ability of a classifier to distinguish between different scenarios of chromosome segregation would be of great interest to the scientific community, including both the fields of molecular dynamics modeling and machine learning. I look forward to the future work hinted at by the authors in which the models presented here could be tested on trajectories obtained from single-particle tracking experiments.

Nevertheless, I recommend some changes to be made before the manuscript can be accepted for publication in PLOS ONE. Please find my comments below.

Major points:

1.) I think it is more sensible to perform cross-validation in the beginning, that is to have repeated splits of the data in training and test sets, then to perform the hyperparameter estimation as described and to average the results. Doing this would ensure that the very high obtained accuracies of the classifiers are not perceived to simply be due to a specific 70/30 split in which training and test data could be similar by chance.

2.) For the hyperparameter tuning, the authors chose k=3 folds for the cross-validation.

It is more typical to perform k-fold cross-validation with a higher number of folds such as k=5 or k=10, see for example (Hastie, T., Tibshirani, R., and Friedman, J. H. The Elements of Statistical Learning. Springer Series in Statistics, 2001) and (James, G., Witten, D., Hastie, T., and Tibshirani, R. An Introduction to Statistical Learning. Springer, 2013).

It would be better to use one of these standard values in addition to the k=3 cross-validation. In case that k=3 was chosen due to the number of 3000 trajectories, then this should be explicitely mentioned. This choice of k and the number of settings should also be highlighted in Fig. 3, not only in the Appendix.

3.) The authors refer to Fig. 8 showing the overall classification accuracy as a function of trajectory length. I could not find this figure in the manuscript.

Minor points:

1.) Gradient Boosting outperforms all other approaches in Tables 3 and 5. This result should be expanded upon some more in the discussion.

2.) References should be checked for proper capitalisation of journal names and article titles.

3.) The authors should refrain from starting sentences with phrases such as „Obviously,…“. Likewise, instead of phrases such as „excellent prediction accuracies“, „slight tendency to overfitting“ or „all classifiers more or less agree“ it is more fitting to limit oneself to presenting numerical values.

4.) Miscellaneous typos in lines 95, 117, 1024, 1055, and 1078.

5.) Missing spaces in lines equations 14 and the caption of Table 2.

6.) The name of used scikit-learn function is missing in line 637.

6. PLOS authors have the option to publish the peer review history of their article (what does this mean?). If published, this will include your full peer review and any attached files.

Reviewer #1: No

Reviewer #2: No

---

## [Author Response · Author response to Decision Letter 0]

1 Nov 2021

Dear Dr. Kestler,

We would like to thank you for the thorough evaluation of our manuscript “Machine Learning Classification of Trajectories from Molecular Dynamics Simulations of Chromosome Segregation” (Submission ID PONE-D-21-29250). We also thank the reviewers for their thoughtful comments and helpful suggestions. We have rewritten the manuscript to address their comments and to include their suggestions. Together with the revised manuscript we enclose a detailed point-by-point response to all recommendations and criticisms of the two reviewers.

Sincerely yours and on behalf of all authors,

Peter Lenz

Point-by-point responses to reviewers' comments: 

Reviewer 1

General comments:

This manuscript is generally well written and argued. The scientific context of this research work is well introduced within the first chapters, while a solid description of the methods and results obtained are provided in the second part. According to the publication criteria of PLOS ONE, I consider this work to be potentially suitable for acceptance, following minor revision.

Please find below my point to point list of comments:

1. Typos:

. Line 61: "... we briefly summarise this replication...".

. Eq. 14: space after "with".

. Line 482: "... modes of segregation: As entropic..." not capital A

We thank the reviewer for the positive evaluation of our manuscript. The typos have been corrected in the revised manuscript.

2. Line 271: "the mother chromosome was connected to the replication factory by additional harmonic springs". -- How? Through every bead in the chromosome or only a subset of those?

We have made the following addition on p. 7, l. 275-282 to describe the process more clearly:

“For this purpose, the two beads closest to the replication factory were connected to it by additional harmonic springs. During each replication step, those two beads were duplicated and then the subsequent beads were reconnected to the replication factory with harmonic bonds. Between the individual replication steps, the beads of the parent chromosome to be duplicated in the following were thus moved to the replication factory by the harmonic spring force, simulating the pulling effect of the factory model.”

3. Line 292: "... two chromosome arms are connected by an additional SMC bond". -- How is this bond modeled?

We have made the following addition on p. 7-8, l. 302-304 for clarification:

“For this the same harmonic spring potential was used for this as for the connection of the beads to each other and to the replication factory. This simplified approach has also been used successfully in [48] to model the linkage of chromosome arms by SMC.”

4. Line 298: "... we followed the procedure from Lim et al. to get an estimate of the elastic force resulting from the dynamics of the singular loci of the chromosome". -- A quick recall of Lim et al. findings, namely that the origin of the segregation strength provided by the ParAB complex is given directly by the intrinsic elastic properties of the chromosome itself, may be useful to the reader for a complete understanding of the method within this work.

We agree with the reviewer that a brief recall of these findings is useful for the reader. We have therefore made the following addition on p. 8, l. 311-320:

“As described above, the DNA-relay model of Lim et al. suggests that the translocation force of the ParAB complex results directly from the intrinsic elastic properties of the chromosome. It is assumed that the DNA-associated ParA-ATP dimers serve as transient tethers that harness the intrinsic dynamics of the chromosome to relay the partition complex from one DNA region to another [6]. Consequently, Lim et al. were able to estimate the expected elastic force of the ParAB complex by tracking the positions of individual loci before the onset of replication and segregation. The same approach can be used in our MD framework to obtain an estimate of the force of the ParAB system within the simulations.”

5. Line 438: "... we created low-dimensional input vectors from a set of statistical features... ". -- Here, did the authors use the original trajectories or the rescaled ones to extract the statistical features?

We have made an addition in the sentence on p.12, l.463 to make this clear:

“In a second approach we created low-dimensional input vectors from a set of statistical features that were calculated from the original trajectories and designed specifically for this purpose.”

6.Line 447, MSD evaluation: what n (not capital) have been used for the evaluation of the MSD parameter from every trajectory, namely for the first statistical feature introduced? (Assumed obviously that this parameter needs then to be varied to evaluate some of the other features).

We thank the reviewer for bringing this point to our attention. We have made it clearer in the text what the role of the parameter n is and to which value we have set it for the individual features on p. 12, l. 474-481:

“Here, the index n defines the lag time. This is the time step considered in the evaluation of the MSD along the trajectory. For example, for n=2, the formula evaluates the MSD of the trajectory between every point and its evolution after two time steps. To calculate our first feature (MSD), we set n=1. Variation of n to n=1,...,N-1 allows to compute MSD curves for different lag times in the following, from which further features (exponent alpha, mean squared displacement ratio) can be obtained as described below.”

7. Line 450: The text describing the role of n in the formula is not completely clear to me in its formulation. I would describe it a bit more, maybe mentioning something like:"the index n indicates the length of the time-step considered in the evaluation of the MSD along the trajectory. For example, for n=2, the formula evaluates the MSD of the trajectory between every point and its evolution after 2 time steps".

We thank the reviewer for this suggestion. We have included this note and added it to the text as seen in point 6 above.

8. Line 690: Though Fig. 8 is supposed to show some of the most relevant results of this work, I have not been able to find it within the provided pdf file. Is it possible that the mentioned figure is missing?

We apologize for the error in the label/upload of the figure. The figure can now be found with the correct label on p. 20, between l. 725 and 726 and has been uploaded accordingly.

9. Line 773: "The results of our simulations indicate that the ParAB-assisted segregation is the most efficient mechanism. -- This have been proved to be true only within your model (therefore I would add "in our model" at the end of the sentence). As you mentioned later, its efficiency could be "an overestimation of the actual translocation effect by ParAB" due to the "implementation of the ParAB system being based on a simplified assumption".

We thank the reviewer for this note and have added it accordingly on p. 22, l. 810.

Reviewer 2

General comments:

In the article „Machine Learning Classification of Trajectories from Molecular Dynamics Simulations of Chromosome Segregation“, Geisel et al. expand on their previous work, presenting various models for the chromosome segregation process in bacteria. The article presents a novel and interesting approach of using results from molecular dynamics simulations to provide the large amount of data required to train a classifier. The simulations include four different segregation mechanisms, each simulated for a ‚track‘ and ‚factory‘ replication model which distinguish whether or not the replisomes are assumed to be located at a fixed position in the center of the cell.

The setup of the model is well-motivated and explained in great detail, as are all steps of the simulation procedure. All assumptions are shown to be in accordance with the literature. The authors also verified that this is the case for the forces acting in their simulation. It is also nicely shown that it is possible to use only a small number of statistical measures as inputs for the classifier instead of high-dimensional vectors, thus reducing the computational demand.

The ability of a classifier to distinguish between different scenarios of chromosome segregation would be of great interest to the scientific community, including both the fields of molecular dynamics modeling and machine learning. I look forward to the future work hinted at by the authors in which the models presented here could be tested on trajectories obtained from single-particle tracking experiments.

Nevertheless, I recommend some changes to be made before the manuscript can be accepted for publication in PLOS ONE. Please find my comments below.

Major points:

1.) I think it is more sensible to perform cross-validation in the beginning, that is to have repeated splits of the data in training and test sets, then to perform the hyperparameter estimation as described and to average the results. Doing this would ensure that the very high obtained accuracies of the classifiers are not perceived to simply be due to a specific 70/30 split in which training and test data could be similar by chance.

The reviewer raises an important point. We therefore performed additional analyses using 5-fold nested cross-validation in order to evaluate the accuracies of the classifiers on various splits of the data. We have appended the results of this analysis in Table S1 (on p.30) and refer to them in the manuscript on p. 9, l. 347-351. Within the analyses we found that the accuracies of the classifiers are not the result of a single favorable split in training and test data, but can also be confirmed in nested cross-validation. We discussed the nested cross-validation analysis in Appendix S4, and compared it to the results of our single split into training and test sets on p. 29, l. 1130-1160.

2.) For the hyperparameter tuning, the authors chose k=3 folds for the cross-validation.

It is more typical to perform k-fold cross-validation with a higher number of folds such as k=5 or k=10, see for example (Hastie, T., Tibshirani, R., and Friedman, J. H. The Elements of Statistical Learning. Springer Series in Statistics, 2001) and (James, G., Witten, D., Hastie, T., and Tibshirani, R. An Introduction to Statistical Learning. Springer, 2013).

It would be better to use one of these standard values in addition to the k=3 cross-validation. In case that k=3 was chosen due to the number of 3000 trajectories, then this should be explicitely mentioned. This choice of k and the number of settings should also be highlighted in Fig. 3, not only in the Appendix.

We thank the reviewer for the references to the literature and accordingly set k=5 for the nested cross-validation mentioned in point 1.) and also added the value of k in Fig. 3 and accordingly adjusted the caption of the figure.

3.) The authors refer to Fig. 8 showing the overall classification accuracy as a function of trajectory length. I could not find this figure in the manuscript.

We apologize for the error in the label/upload of the figure. The figure can now be found with the correct label on p. 20, between l. 725 and 726 and has been uploaded accordingly.

Minor points:

1.) Gradient Boosting outperforms all other approaches in Tables 3 and 5. This result should be expanded upon some more in the discussion.

As suggested by the reviewer, we have made the following additions on p. 24, l.895-896:

“However, the highest prediction accuracy on both test and training data is achieved with the GB classifier.”

And further down on p. 24, l. 915-919:

“Overall, also for the case of low-dimensional input features, it turns out that the best prediction accuracy is achieved with the GB classifier. The iterative approach of the GB classifier to gradually train new decision trees based on the previously made errors of the ensemble thus turns out to be the most successful algorithm for classifying the segregation mechanisms in our study.”

2.) References should be checked for proper capitalisation of journal names and article titles.

We have checked and adjusted the references for correct capitalization of journal names and articles.

3.) The authors should refrain from starting sentences with phrases such as „Obviously,…“. Likewise, instead of phrases such as „excellent prediction accuracies“, „slight tendency to overfitting“ or „all classifiers more or less agree“ it is more fitting to limit oneself to presenting numerical values.

We have accordingly changed the text to reduce in the following lines:

- p. 15, l.564

- p. 15, l. 565

- p. 15, l.567

- p. 16, l. 596

- p. 19, l. 690-691

4.) Miscellaneous typos in lines 95, 117, 1024, 1055, and 1078.

We have corrected the typos and thank the reviewer for bringing them to our attention.

5.) Missing spaces in lines equations 14 and the caption of Table 2.

We have added the spaces.

6.) The name of used scikit-learn function is missing in line 637.

The name was added to the text on p. 19, l. 671-672.

Additional changes: 

- We have corrected typos on p. 17, l. 616 and p.31, l. 1212-1216.

- We have adjusted the capitalization of the headings, citations of the figures and tables as well as the author affiliations to the style requirements.

- We have removed the funding information from the acknowledgments as requested by the Editor.

---

## [Decision Letter · Decision Letter 1]

25 Nov 2021

PONE-D-21-29250R1Machine Learning Classification of Trajectories from Molecular Dynamics Simulations of Chromosome SegregationPLOS ONE

Dear Dr. Lenz,

Thank you for submitting your manuscript to PLOS ONE. After careful consideration, we feel that it has merit but does not fully meet PLOS ONE’s publication criteria as it currently stands. Therefore, we invite you to submit a revised version of the manuscript that addresses the remaining points raised during the review process.

We look forward to receiving your revised manuscript.

Kind regards,

Hans A. Kestler

Academic Editor

PLOS ONE

Journal Requirements:

Reviewers' comments:

Reviewer's Responses to Questions

**Comments to the Author**

1. If the authors have adequately addressed your comments raised in a previous round of review and you feel that this manuscript is now acceptable for publication, you may indicate that here to bypass the “Comments to the Author” section, enter your conflict of interest statement in the “Confidential to Editor” section, and submit your "Accept" recommendation.

Reviewer #1: (No Response)

Reviewer #2: All comments have been addressed

2. Is the manuscript technically sound, and do the data support the conclusions?

Reviewer #1: Yes

Reviewer #2: Yes

3. Has the statistical analysis been performed appropriately and rigorously? 

Reviewer #1: Yes

Reviewer #2: Yes

4. Have the authors made all data underlying the findings in their manuscript fully available?

Reviewer #1: Yes

Reviewer #2: Yes

5. Is the manuscript presented in an intelligible fashion and written in standard English?

Reviewer #1: Yes

Reviewer #2: Yes

6. Review Comments to the Author

Reviewer #1: The authors have addressed adequately and thoroughly all of my concerns raised in the previous round of review. I will therefore recommend this work for publication after a final comment on Figure 8, which has been added in this latest version of the paper. For a clear visualization of the results, I think it would be helpful for the authors to provide an enlarged view of the very first part of the graph, showing the "Accuracy" for the values of the parameter "s" approximately in the range (0, 125), with labels showing the values of "s" for each data point entered, as an extra figure or within the figure itself. From the text I could only see that the very first point is evaluated for s = 5, but I'd like to know what other values were evaluated for this analysis, and better visualize how the "Accuracy" scales as the value of "s" increases in this very interesting range.

Reviewer #2: (No Response)

7. PLOS authors have the option to publish the peer review history of their article (what does this mean?). If published, this will include your full peer review and any attached files.

Reviewer #1: No

Reviewer #2: No

---

## [Author Response · Author response to Decision Letter 1]

2 Dec 2021

Dear Dr. Kestler,

We would like to thank you for the thorough evaluation of our manuscript “Machine Learning Classification of Trajectories from Molecular Dynamics Simulations of Chromosome Segregation” (Submission ID PONE-D-21-29250). We also thank the reviewers for their re-reading of our manuscript and the final helpful comment to improve our figure. We have addressed this last comment in the revised manuscript and described the changes we made in the detailed point-by-point response to the reviewers.

Sincerely yours and on behalf of all authors,

Peter Lenz

Point-by-point responses to reviewers' comments: 

Reviewer 1

General comments:

The authors have addressed adequately and thoroughly all of my concerns raised in the previous round of review. I will therefore recommend this work for publication after a final comment on Figure 8, which has been added in this latest version of the paper. For a clear visualization of the results, I think it would be helpful for the authors to provide an enlarged view of the very first part of the graph, showing the "Accuracy" for the values of the parameter "s" approximately in the range (0, 125), with labels showing the values of "s" for each data point entered, as an extra figure or within the figure itself. From the text I could only see that the very first point is evaluated for s = 5, but I'd like to know what other values were evaluated for this analysis, and better visualize how the "Accuracy" scales as the value of "s" increases in this very interesting range.

We thank the reviewer for the positive evaluation of our manuscript. As suggested, we added a subplot to Figure 8 which shows an enlarged view of the first part of the graph for a better visualization of the results. We have also made this clear in the caption of the figure on p. 20 between l. 716-717 by adding the following sentence:

“The subplot provides an enlarged view of the very first part of the graph with the length of the trajectories ranging from 5s to 100s.”

Reviewer 2

Reviewer 2 stated that all comments were addressed and had no further requests for improvement.

---

## [Editor Report · Decision Letter 2]

19 Dec 2021

Machine Learning Classification of Trajectories from Molecular Dynamics Simulations of Chromosome Segregation

PONE-D-21-29250R2

Dear Dr. Lenz,

We’re pleased to inform you that your manuscript has been judged scientifically suitable for publication and will be formally accepted for publication once it meets all outstanding technical requirements.

Kind regards,

Hans A. Kestler

Academic Editor

PLOS ONE
---

## [Editor Report · Acceptance letter]

11 Jan 2022

PONE-D-21-29250R2 

Machine Learning Classification of Trajectories from Molecular Dynamics Simulations of Chromosome Segregation  

Dear Dr. Lenz:

I'm pleased to inform you that your manuscript has been deemed suitable for publication in PLOS ONE. Congratulations! Your manuscript is now with our production department. 

Kind regards, 

on behalf of

Prof. Hans A. Kestler 

Academic Editor

PLOS ONE